# Strain-driven domain wall network with chiral junctions in an antiferromagnet

Vishesh Saxena[1], Mara Gutzeit[2], Arturo Rodríguez-Sota[1], Soumyajyoti Haldar [2], Felix Zahner [1], Roland Wiesendanger [1], André Kubetzka[1], Stefan Heinze [2,3] & Kirsten von Bergmann [1] ✉

Antiferromagnetic materials have recently emerged as promising candidates in spintronics. At the same time, more complex localized non-coplanar magnetic states such as skyrmions are in the research focus due to their intriguing dynamical and transport properties. Recently, a conceptual shift has occurred to envision the use of such magnetic defects not only in one-dimensional race track devices but also to exploit their unique properties in two-dimensional networks. Here we use local strain in a collinear antiferromagnetic film to induce a complex domain wall network. Using spin-polarized scanning tunneling microscopy we characterize the different building blocks of the network – ranging from collinear magnetic domains, over non-collinear domain walls, to non-coplanar localized domain wall junctions – on the atomic scale. We find that the triple domain wall junctions exhibit a structural handedness. The origin is an exchange-driven lateral relaxation as explained using first-principles calculations. We predict that the domain wall junctions exhibit topological orbital magnetization generated by their non-coplanar spin structure, implying topological transport properties due to the network.

Antiferromagnets have moved into the focus of application-related research due to their favorable properties such as ultrafast spin dynamics, lack of stray fields, and abundance in various material classes[1–3]. In addition to collinear antiferromagnets, also systems with non-collinear magnetic order as in $Mn_3Sn$[4,5] or non-coplanar order as in $CoTa_3S_6$[6,7] have attracted significant attention in view of future antiferromagnetic spintronics devices. At the same time, the absence of a net magnetization and the local compensation of magnetic moments on the atomic scale make experimental investigations difficult[8,9].

In addition to the direction of the Néel vector and the size of domains, the properties of antiferromagnetic domain walls are also relevant for spintronics applications. The experimental characterization of the internal spin structure of antiferromagnetic domain walls is challenging and has been realized mostly by spin-polarized scanning tunneling microscopy (SP-STM)[10–12] or scanning nitrogen-vacancy (NV) center microscopy techniques[13,14]. For phase-shifted

antiferromagnetic domains, the transition was found to be a continuous rotation of the Néel vector across the domain wall[10]. Both Bloch- and Néel-type rotations, as well as intermediate domain walls were proposed based on the emerging stray field that has been measured by an NV center[13].

In contrast, orientational antiferromagnetic domain walls were found to be fundamentally different compared to ferromagnetic domain walls, as the lowest energy state can be accompanied by orthogonal spin configurations in the center of the wall, namely so-called superposition walls where the adjacent antiferromagnetic domains mix and fade out across the wall[11]. This configuration arises in antiferromagnets where the ground state can be described by a spin spiral (1Q) state, which has at least two symmetry-equivalent orientational domains, e.g., a row-wise antiferromagnetic (RW-AFM) on a square or hexagonal atomic lattice. In the absence of higher-order interactions this RW-AFM (1Q) state is degenerate with the

[1]Institute of Nanostructure and Solid State Physics, University of Hamburg, Hamburg, Germany. [2]Institute of Theoretical Physics and Astrophysics, University of Kiel, Kiel, Germany. [3]Kiel Nano, Surface, and Interface Science (KiNSIS), University of Kiel, Kiel, Germany. ✉e-mail: kirsten.von.bergmann@physik.uni-hamburg.de

corresponding 2Q or 3Q superposition states formed by combining their symmetry-equivalent rotated 1Q-states.

Whereas in collinear antiferromagnets with one preferred axis only one type of domain wall occurs which separates the two possible distinct phase-shifted domains[15], there are also materials that can host different types of domain walls and also junctions thereof. Recently, it has been shown that a disordered network of antiferromagnetic domain walls can form with specific junctions that were characterized with respect to their topological complex charges[12]. Such domain walls and junctions can be classified as extended or localized magnetic defects. Also in other magnetic systems such magnetic defects have been investigated, for instance in lamellar stripe domains where topological domain walls were found[16,17]. In the case of three preferred directions of spin spiral propagation, different junction-types have been observed and characterized at points where three orientational domains meet[16,18–20]. Often such localized magnetic objects show non-coplanar spin textures, similar to skyrmions that have been in the focus of recent research[21]. One intriguing property of non-coplanar spin textures is their response to spin or charge currents, which can manifest in motion of particle-like states[22], formation of topological orbital moments[23], and contributions to topological Hall signals[6,7,24]. Such localized magnetic objects, extended defects, or junctions can form networks, and in the form of magnetic skyrmions they have been investigated in the context of novel computing schemes as neuromorphic or reservoir computing[25–28].

In a spin spiral phase the position of a localized magnetic junction was found to be associated with a curved sample surface and strain has been proposed as driving force for their formation[19]. A recent study has deliberately introduced strain to impose the wave vector orientation of adjacent stripe domains with a period on the several-micrometer scale[29]. A large-scale patterning of stripe domains that exhibit a period of about half a micron was realized and demonstrated for various geometries. A significant magneto-elastic energy is also present in antiferromagnets, which manifest in, for instance, structural distortions below the Néel temperature[30,31], a shape-induced anisotropy of the Néel vector orientation[32], or a preferred domain structure[33,34]. Also, the properties of non-collinear antiferromagnets were found to be highly sensitive to strain[35]. Recently, strain-related control of antiferromagnets has evolved as a new avenue for potential applications in spintronics[36–39].

Here, we use SP-STM to reveal the building blocks of a complex strain-induced domain wall network in an antiferromagnetic Mn film on the atomic scale, ranging from collinear magnetic domains, over non-collinear domain walls, to non-coplanar localized domain wall junctions. Two types of triple domain wall junctions are observed which show structural chirality. This results from a surprising lateral shift of the top magnetic layer due to strong antiferromagnetic exchange forces as explained based on density functional theory (DFT). This shearing of the film generates local strain at the domain walls. We further demonstrate that the domain wall junctions exhibit a non-coplanar spin structure—the so-called 3Q state—which exhibits an emergent localized topological orbital magnetization. Such topological orbital moments give rise to contributions to the anomalous Hall effect, in addition to the expected complex transport characteristic of a network in which collinear, non-collinear and non-coplanar magnetic regions coexist.

## Results

### Building blocks of the domain wall network

We study an ultra-thin film system that hosts a row-wise antiferromagnetic (RW-AFM) state on a hexagonal lattice, namely a two atomic layer thick Mn film on the (111) surface of an Ir crystal (see "Methods"). In this system, the RW-AFM state has three symmetry-equivalent orientational domains. An overview STM constant-current image is presented in Fig. 1a. The bright lines that form a network are domain walls separating adjacent orientational domains of the RW-AFM state. The bright spots on the order of 5–10 nm in diameter and 15–130 pm in height are Ar bubbles located below the surface[40] (see also Supplementary Fig. S1). The domain wall network can also be seen in the simultaneously acquired map of differential conductance (dI/dU) in Fig. 1b. Here, the Ar bubbles are nearly invisible, and the domain walls appear as dark lines.

A closer view of a domain wall between two orientational RW-AFM domains is presented in Fig. 1c: for this SP-STM constant-current image a magnetic tip was used, and the resulting tunnel current scales with the projection of tip and sample magnetization (see "Methods"). The RW-AFM state is identified in SP-STM images by its characteristic appearance of alternating bright and dark lines, which correspond to atomic rows of opposite spin directions[41,42], see also the to-scale

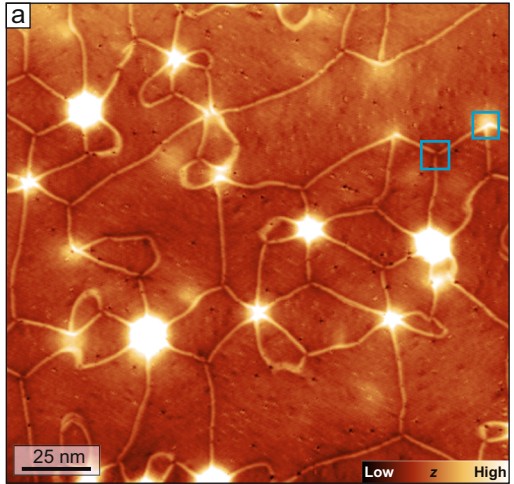

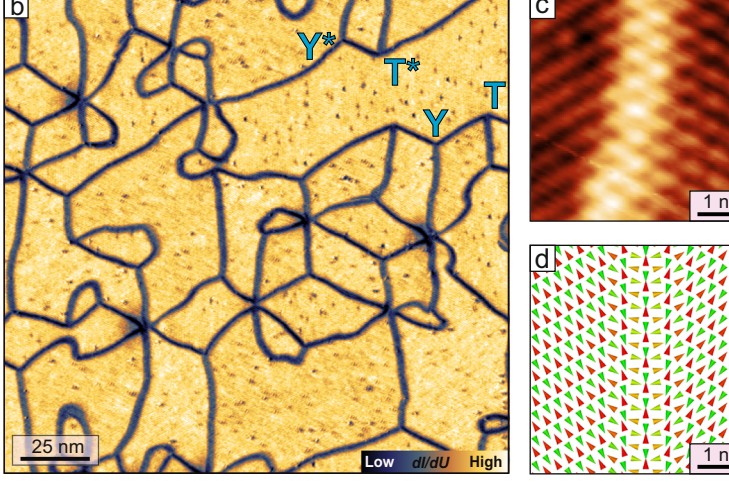

**Fig. 1 | Strain-induced domain wall network in a collinear antiferromagnet.**
**a** Constant-current STM image of a Mn double-layer film on Ir(111); the bright spots correspond to Ar bubbles below the Ir surface; bright lines indicate domain walls between orientational antiferromagnetic domains ($\Delta z = 60$ pm); cyan boxes indicate sample areas which are shown again in Fig. 2c, d. **b** dI/dU map acquired simultaneously with (**a**), labels refer to the types of triple-junction. **c** SP-STM constant-current image of a domain wall between two orientational antiferromagnetic domains ($\Delta z = 15$ pm). **d** Spin model of a superposition wall between orientational antiferromagnetic domains (see "Methods"); cones indicate atomic magnetic moments, red and green indicate opposite vertical in-plane magnetization components, while yellow indicates horizontal in-plane magnetization components. (Measurement parameters: **a**, **b** $U = +10$ mV, $I = 5$ nA; **c** $U = +10$ mV, $I = 3$ nA; all: $T = 4$ K).

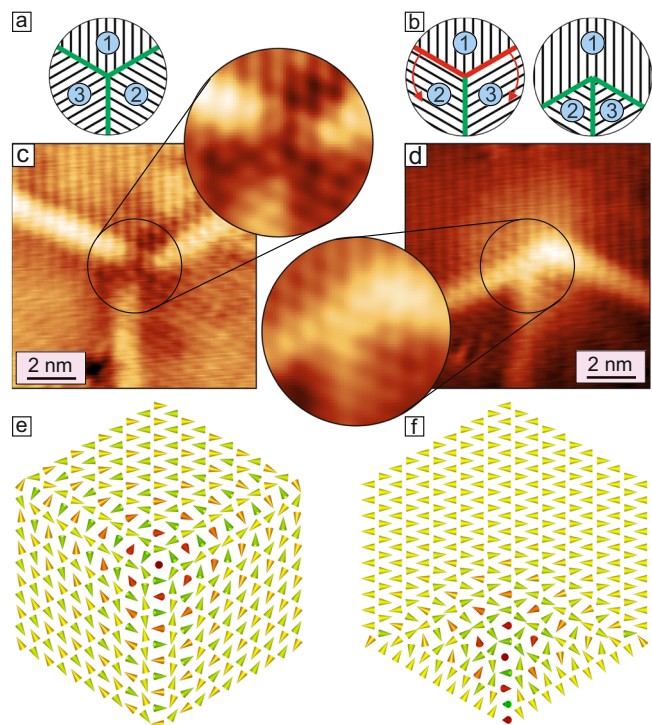

**Fig. 2 | Domain wall triple-junctions. a, b** Schematics of the two different types of triple junctions, i.e., Y and T, respectively; numbers refer to the three possible orientational antiferromagnetic domains, green lines indicate the favorable 120° domain walls. **c, d** SP-STM constant-current images of a Y- and a T-junction, respectively ($\Delta z = 36$ pm and $\Delta z = 25$ pm), see boxes in Fig. 1a for their positions within the network. The insets show a magnified view of the central area of the junction exhibiting the hexagonal magnetic pattern characteristic for the 3Q state; for better visibility of the magnetic pattern here high frequency noise has been removed by a lowpass filter with a cut-off frequency corresponding to a wavelength of 0.4 nm. ($U = +10$ mV, $I = 5$ nA, $T = 4$ K). **e, f** Spin models of the two types of junctions (see "Methods"); cones indicate atomic magnetic moments, red and green indicate opposite out-of-plane magnetization components, while yellow indicates in-plane magnetization components; models are not to scale.

illustration of the magnetic state in Fig. 1d. The lines visible in the two orientational domains of Fig. 1c, d enclose an angle of 120°, and we find that all of the straight walls in the overview image (Fig. 1a, b) are of this type. Towards the center of this domain wall the lines transition into a dotted pattern which locally has hexagonal symmetry. Thus, we conclude that they are 2Q-superposition domain walls as previously observed for the RW-AFM state in the Mn monolayer on Re(0001)[11]. These walls represent a continuous transition between adjacent orientational domains via a superposition state that is characterized by several 90° angles between neighboring moments, see spin dynamics simulation of such a wall in Fig. 1d with generic model parameters (see "Methods").

Within the network, three symmetry-equivalent orientational domains of the RW-AFM state exist. When all three of them meet in one point triple-domain-wall-junctions are formed. The network consists of different connections of domain walls, and two different types of triple domain wall junctions are found as indicated in Fig. 1b: either the triple-junction has threefold rotational symmetry, and we will refer to them as Y/Y*-junctions, or the threefold symmetry is broken, and we call them T/T*-junctions from now on. These types of junctions are reminiscent of some of those previously observed in lamellar stripe domains[16,18–20], and we can rationalize their structure by the sketches in Fig. 2a, b: when all three possible orientational domains meet in one point they can have two different configurations, as indicated by the order of the numbers. Either the numbers increase when one goes

around the center of the junction in a clockwise fashion, see Fig. 2a, or one has to go around in an anticlockwise fashion, see Fig. 2b. With the constraint that all straight walls are 120° walls (indicated by green lines) we see that only the clockwise configuration can have a symmetric Y-junction, while the anticlockwise configuration has to adapt its wall path to form a more asymmetric T-shape.

Closer-view SP-STM constant-current images of a Y- and a T-junction are displayed in Fig. 2c, d and in both configurations the characteristic 120° domain walls are seen. We find that, depending on the specific magnetization direction of the employed tip, the hexagonal magnetic patterns of the three walls merge in the center of the junctions (see also Fig. S2). The smooth connection of these 2Q domain walls suggests that in the center a 3Q state forms, which contains components of all 1Q states. We would like to note that all triple-junctions in the network displayed in Fig. 1 have these characteristic patterns. The corresponding sketches in Fig. 2e, f are the result of spin dynamics simulations and show that indeed at the center of the junctions a 3Q state emerges, which is the superposition state of all three adjacent antiferromagnetic domains. This 3Q state, or triple-Q state, is a nano-scale non-coplanar spin state characterized by tetrahedron angles between all nearest neighbors and has four atoms in the magnetic unit cell[6,7,42–44].

## First-principles calculations

To obtain deeper insight into the magnetic properties of this system, we employed DFT (see "Methods" for computational details). We have performed total energy calculations for various collinear and non-collinear magnetic states of the Mn double-layer on Ir(111). To scan a large part of the magnetic phase space we have calculated for a pseudomorphic Mn double-layer on Ir(111) the energy dispersion of spin spiral (1Q) states[45,46] which constitute the general solution of the Heisenberg model on a periodic lattice. Within the Heisenberg model of pair-wise exchange, superpositions of symmetry equivalent spin spirals— so-called multi-Q states—are energetically degenerate with the corresponding spin spiral states. However, higher-order exchange interactions such as biquadratic and four-spin terms[42,43,47,48] can lift this degeneracy. These interactions arise from the Hubbard model in fourth-order perturbation theory[47,49,50]. Therefore, they are typically much smaller than the pair-wise Heisenberg exchange which occurs in second order. Note, that within DFT all the magnetic interactions are implicitly included within the exchange-correlation functional and contribute to the total energy of a considered magnetic state.

We have considered the 3Q state (Fig. 3a, b) that is a superposition of the three RW-AFM (1Q) states (Fig. 3c, d) propagating along the equivalent crystallographic directions of the hexagonal 2D unit cell. In a double-layer, the 3Q state can occur in two different versions, one with a more antiparallel alignment between neighboring spins of the two layers ($3Q_{\rightleftarrows}$), and one with a more ferromagnetic alignment ($3Q_{\rightrightarrows}$), see Fig. 3a,b, respectively. We find that the $3Q_{\rightleftarrows}$ state has a significantly lower energy, implying an antiferromagnetic exchange coupling between the Mn layers, i.e., $H = -J_{BT}(\mathbf{s}_B \cdot \mathbf{s}_T)$ where $\mathbf{s}_B$ and $\mathbf{s}_T$ represent nearest neighbor (NN) spins of the bottom (B) and top (T) layer, respectively, and $J_{BT} < 0$ is the exchange constant. Note that the interlayer exchange constants have been obtained via DFT based on spin spiral calculations[45,46].

For the 1Q states, we have also considered both antiferro- and ferromagnetic configurations between the layers. In the RW-AFM$_{\rightleftarrows}$ state—the energetically lowest 1Q state obtained in the DFT calculations[46]—two of the three NN Mn atoms of the bottom layer exhibit a spin direction antiparallel to that of the top layer Mn atom (see blue triangle in Fig. 3c). This spin alignment is energetically favored by the antiferromagnetic exchange coupling between the Mn layers. In contrast, the RW-AFM$_{\rightrightarrows}$ state with a net ferromagnetic alignment ($\rightrightarrows$) is unfavorable since for a given Mn top layer atom two of the three NN spins in the bottom Mn layer are parallel (see blue triangle

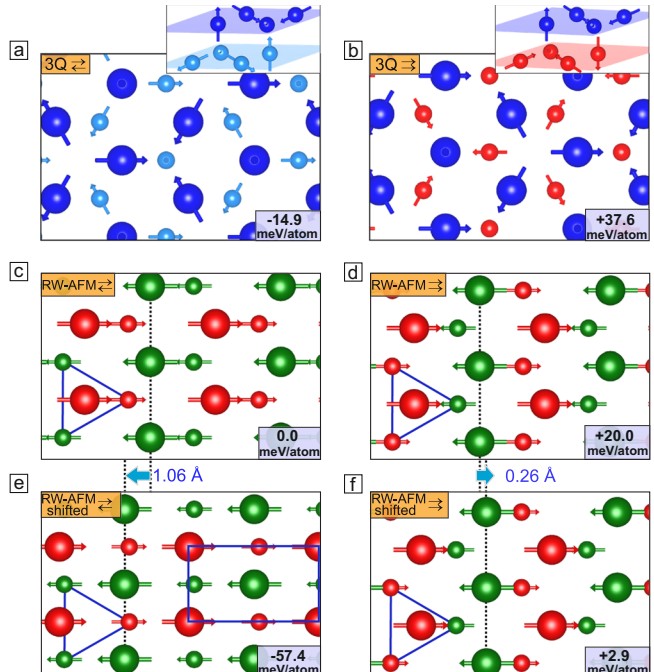

**Fig. 3 | Competing spin configurations of the Mn double-layer on Ir(111).**
**a, b** Top view of the 3Q state for a net AFM (⇄) and FM (⇉) coupling between the moments of the Mn layers. The top layer atoms are represented by larger spheres, see also insets for perspective views. **c, d** Top view of the corresponding RW-AFM states; each of them represents one orientation of the 1Q states that lead to the 3Q superposition states in (**a, b**); blue triangles indicate the triangular lattice of the lower Mn layer. **e, f** Laterally relaxed RW-AFM states, where the displacement of the Mn atoms in the top layer relative to the hollow sites is indicated by cyan arrows. The total energies of each magnetic structure are given with respect to the RW-AFM ⇄ state.

in Fig. 3d). Comparing the DFT total energy calculations, we find that the 3Q⇄ state (Fig. 3a) is energetically favored over the corresponding single-Q RW-AFM⇄ state (Fig. 3c) by about 15 meV/Mn atom. This indicates a significant effect of higher-order interactions which can be revealed explicitly by mapping the DFT total energies to an atomistic spin model of the double layer[46].

In the experimental study, the RW-AFM state is present in the domains and the 3Q state is only visible in the junctions. This is in contradiction to the DFT result that the 3Q⇄ state (Fig. 3a) has the lowest DFT total energy. However, so far we have considered a pseudomorphic Mn double-layer on the Ir(111) surface in the DFT calculations and not yet taken into account that the RW-AFM state breaks the threefold symmetry of the substrate. In the system displayed in Fig. 3c only one mirror plane perpendicular to the dotted line remains. As long as this symmetry is maintained, the Mn atoms may in principle shift out of the hollow sites, which may lead to a lower total energy of this magnetic state[51]. Indeed, upon relaxing the lateral coordinates of the Mn atoms within the RW-AFM⇄ state (Fig. 3c) in the DFT calculations, we find a large shift of the top Mn layer by about 1 Å along the [11$\bar{2}$]-direction. This shearing of the film moves the Mn atoms of the top layer nearly into the bridge-site positions (Fig. 3e) and leads to a significant total energy reduction of about 57 meV/Mn atom. As a result, the sheared RW-AFM⇄ state is about 42 meV/Mn atom lower than the 3Q⇄ state (Fig. 3a) and becomes the energetically favored state within DFT.

The origin of the surprising structural shearing in the RW-AFM⇄ state is the antiferromagnetic exchange interaction between NN Mn atoms of top and bottom layer. The lateral shift of the top Mn layer moves the top layer Mn atoms closer to the two NN bottom layer Mn atoms with favorable antiparallel spin alignment (see blue triangle in

Fig. 3e) and increases the distance to the bottom layer Mn atom with the unfavorable parallel spin alignment. Thereby, the antiferromagnetic pair-wise exchange interaction between NN top and bottom layer Mn spins, $J_{BT}$, is the driving force for the shift because it lowers the energy of the sheared RW-AFM⇄ state. The optimization of the pair-wise exchange energy in the sheared RW-AFM⇄ state further overcompensates the small energy gain due to higher-order exchange in the 3Q⇄ state.

Note, that the RW-AFM⇉ structure with its net FM coupling of the moments between the two layers (blue triangle in Fig. 3d) also experiences a lateral shift (Fig. 3f). However, the effect is much less pronounced with the Mn atoms of the top layer being displaced by only 0.26 Å (Fig. 3d). Accordingly, there is a much smaller energy gain of about 17 meV/Mn atom with respect to the unshifted state, i.e., the hollow-site stacking. This can be explained by the fact that only one of the NN Mn atoms of the bottom layer is antiparallel (see blue triangle in Fig. 3f), and the layer can only move slightly to increase the antiferromagnetic interlayer coupling, because a larger shift would lead to an unfavorable on-top adsorption site. For the FM and layered AFM state, no lateral shift occurs in our DFT calculations which further supports the conclusion of an exchange-driven mechanism (see Fig. S3 and Table S4).

## Characterization and chirality of the junctions

Based on these DFT calculations, we can explain the experimental observation of RW-AFM domains in the SP-STM images and predict a large lateral shift of the top Mn layer in the domains. For the two magnetic states of lowest energy, i.e., the 3Q⇄ state and the shifted RW-AFM⇄ state (Fig. 3a, e), we have calculated spin-averaged STM images based on the Tersoff-Hamann model[52] (see "Methods"). Figure 4a, b shows that in both cases we find anticorrugation, i.e., at the position of the surface atoms the signal is minimal[53]. Whereas the image of the 3Q⇄ state in Fig. 4a shows threefold symmetry, the image of the shifted RW-AFM⇄ in Fig. 4b reflects the twofold symmetry of the uniaxial magnetic state, even in the displayed spin-averaged case.

Figure 4c shows an atomically resolved constant-current STM image of a Y*-junction, and we find that also here the pattern changes between the center of the junction and the extended RW-AFM domains. Assuming that the positions of lowest signal indicate the atom positions, as seen in the DFT calculations, we find by extrapolation of the atomic rows in the RW-AFM domains (see black line) that the top-layer Mn atoms are not aligned across adjacent orientational domains, providing experimental evidence for the lateral shear of the Mn film predicted by DFT, see cyan arrows.

The lateral shearing has important consequences for the structural environment of the atoms at the domain walls. A simplified model of the shifted top Mn layer of the three domains around a Y* junction is displayed in Fig. 4d, where yellow and red indicate perfect bridge and hollow site positions, respectively. It can be seen that the atoms of the three orientational domains are shifted in different directions (cyan arrows). As atoms are forced to shift from one bridge site to a different bridge site when going from one domain to another, there is a variation of the atom-atom distances, leading to local strain at the domain wall. In a theoretical study of a system with orthogonal orientational domains, it has been found that the magneto-elastic energy present at domain walls strongly couples their path to the orientation of the adjacent domains[34], a mechanism that might also be responsible for the strong preference for straight 120° walls in the system studied here.

Figure 5 shows part of a network with several domain wall junctions. Close inspection of the triple-junctions shows that the shear of the Mn film is also reflected by a shift of the apparent domain wall positions: when the straight 120° walls are extrapolated to the junction's center, they do not merge in one point but instead span a triangle, as evident in the enlarged views at the top. This results in a

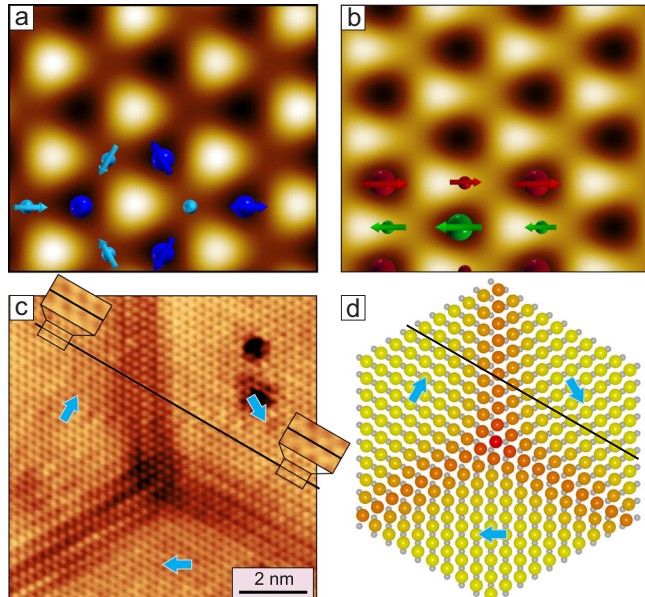

**Fig. 4 | Magnetism-driven structural shift. a** STM image calculated via DFT for the 3Q⇌state (Fig. 3a). Large (small) spheres indicate atoms of the top (bottom) Mn layer. **b** STM image calculated via DFT for the shifted RW-AFM⇌state (Fig. 3e). (For an STM image calculated via DFT for the unshifted RW-AFM⇌state (Fig. 3c) see Fig. S4). **c** Atomic resolution constant-current STM image of a Y*-junction; the black line is placed across an atomic row (dark spots) in the right domain, and extrapolation to the top domain confirms the different relative shifts for the top Mn layer, as indicated by cyan arrows ($\Delta z = 25$ pm, $U = +10$ mV, $I = 5$ nA, $T = 4$ K). **d** Atomistic model of a Y*-junction where the atom positions vary from hollow-site in the center (3Q) to bridge site in the RW-AFM domains; the color code illustrates the size of the shift (red: no shift and yellow: shift to bridge site), which leads to a structural handedness with compressive strain on one side and tensile strain on the other side of each domain wall.

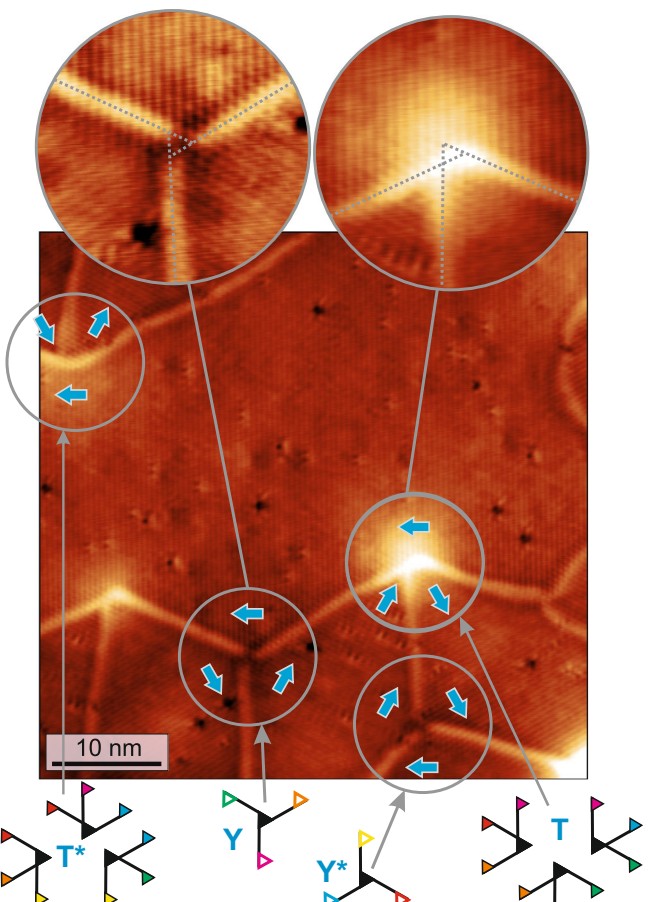

**Fig. 5 | Chirality of the triple-junctions.** SP-STM constant-current image of several triple-junctions, the different types are labelled; cyan arrows indicate the shift direction of the top Mn layer for the different orientational domains ($\Delta z = 50$ pm). Drawing lines along the 120° domain walls in different junctions results in an asymmetric intersection (see sketches) of the three domain walls due to the structural shift. Sketches show all possible triple-junctions that can occur with 120° domain walls, the T/T*-junctions occur in three orientations. To form a network, a pair of empty (Y/Y*-junction) and filled same color triangles (T/T*-junction) need to be connected, i.e., any straight wall must have a Y/Y*-junction at one end and a T/T*-junction at the other end respectively which is the only possible way of forming the network. (Measurement parameters: $U = +10$ mV, $I = 5$ nA, $T = 4$ K).

windmill-like shape for the Y-junctions and an asymmetric appearance of the T-junctions. We find that also all other Y- and T-junctions show the same characteristics. This structural chirality of the junctions is linked to the shear directions of the adjacent domains (see cyan arrows). Our analysis shows that Y-junctions and Y*-junctions have opposite structural chirality, in line with their opposite radial shift direction of the Mn top layer in the adjacent domains (cyan arrows). Also the structures of the asymmetric T- and T*-junctions are connected by mirror symmetry. Due to their lower symmetry the T/T*-junctions can occur in three orientations. Sketches of all possible triple-junctions are shown at the bottom of Fig. 5.

With regards to the domain wall network, it is worth to note that due to the unique domain configurations around these triple-junctions they always come in pairs of Y(*) and T(*). This can be understood from the arrangement of the three orientational RW-AFM domains around the triple-junctions center. As seen in Fig. 2a, b, the order of the numbered orientational domains is clockwise for Y(*)-junctions, whereas it is anticlockwise for all T(*)-junctions. The constraint to always pair Y(*) and T(*) junctions can be regarded as an analogy to e.g., (anti)merons pairing. In that case opposite helicity, which describes the in-plane magnetization arrangement around the core, is needed to form a pair[54,55]. The possible pairing configurations for the Y(*)- and T(*)-junctions can be seen in the sketches at the bottom of Fig. 5, where a connection of junctions can only happen between open and filled triangles of the same color.

## Origin of the network
After the analysis of the magnetic state and the structural chirality in the triple-junctions, we turn to the question why so many domain walls

appear in the first place. While in ferromagnets the incorporation of domain walls can lower the total energy due to a reduction of the stray field contributions, in antiferromagnets this is not the case, and domain walls in ultra-thin films always have a positive energy contribution. We find that in our sample the Ar bubbles connect up to six walls, see constant-current images shown in Figs. 1a and 6a, and conclude that they initiate the formation of the observed domain wall network. Indeed, samples without Ar bubbles do not show such a high density of domain walls (see Fig. S1). Due to the comparably large height of the Ar bubble, the magnetic pattern of the hexa-junction can be better seen in the simultaneously acquired current map, as shown in Fig. 6b for the left hexa-junction of Fig. 6a. Again, the pattern in the junction center is hexagonal, indicative of a local 3Q state. The orientational RW-AFM domains around the Ar bubbles always follow the same sequence, where the rows of parallel spins are always tangential. Such a configuration with six domain walls is also illustrated in the spin model of Fig. 6c (see "Methods"). The shear direction around the Ar bubble is indicated by the cyan arrows in Fig. 6a. The experimentally observed offset of the wall positions with respect to the center of the junction reflects the strain due to this shearing of the Mn film. The

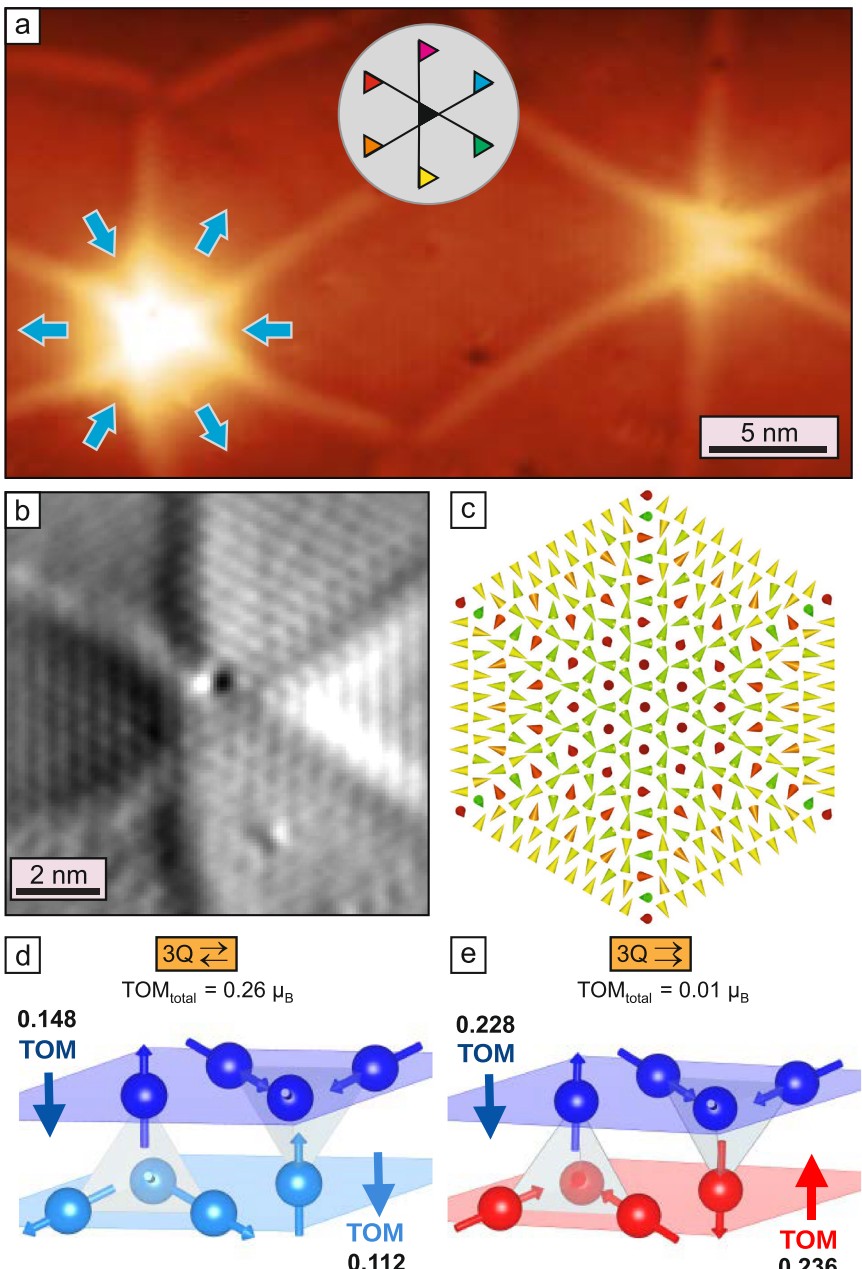

**Fig. 6 | Hexa-junctions and topological orbital moments. a** SP-STM constant-current image of the antiferromagnetic domain wall network with two hexa-junctions located at Ar bubbles ($\Delta z = 150$ pm), cyan arrows indicate the shift direction of the top Mn layer. **b** Simultaneously measured current map of the left hexa-junction of (**a**), showing the hexagonal magnetic pattern in the junction center indicative of the 3Q state ($\Delta I = \pm 200$ pA) (Measurement parameters: $U = +10$ mV, $I = 1$ nA, $T = 4$ K). **c** Spin model of a hexa-junction; cones indicate atomic magnetic moments, red and green indicate opposite out-of-plane magnetization components, while yellow indicates in-plane magnetization components; model is not to scale. **d**, **e** Sketches illustrating the two possible 3Q configurations for a double-layer (cf. Fig. 3a, b); the arrows indicate the DFT calculated direction of the TOM for each individual layer; values are the DFT calculated layer-resolved and total TOMs for each configuration, and are given in $\mu_B$/magnetic unit cell.

symmetric arrangement of shear directions around the junction suggests a non-sheared locally pseudomorphic Mn double-layer in the Ar bubble center. While this hexa-junction appears to be very symmetric, the majority of the experimentally found hexa-junctions are better characterized as pairs of T/T$^*$-junctions (see for instance the right hexa-junction in Fig. 6a) (see also Fig. S5). Indeed, also single T$^{(*)}$-junctions are typically found at smaller Ar bubbles, whereas the formation of Y$^{(*)}$-junctions is not related to the Ar bubbles (see Fig. S6).

In our double-layer system, the RW-AFM state leads to a shearing of the layer. In turn, a modification of the atomic structure is expected to influence the magnetic state. It has been reported that Ar bubbles

below the surface of metallic single crystals induce a local curvature in the surface layers, which leads to tensile and compressive strain at the top and at the rim of the Ar bubbles, respectively[40] (see Fig. S7). Thus, in the vicinity of the Ar bubbles, the atoms in the Mn double-layer shift due to this local strain. This locally breaks the threefold symmetry at the slopes of the Ar bubble. We speculate that upon cooling through the critical magnetic temperature this local strain field triggers the formation of specific orientational domains on the sides of the Ar bubble, similar to what has been reported before for a preferential orientation of a uniaxial adsorbate arrangement around an Argon bubble[40,56]. This interpretation is supported by our DFT calculations

for the Mn DL on Ir(111) including lateral strain (see Fig. S7). The calculations show that for the radial compressive strain expected at the rim of the Ar bubble, the parallel spin rows of the RW-AFM state prefer to be perpendicular to the strain direction, i.e., tangential with respect to the Ar bubble center, in agreement with the experimental observations. Note that also the single T$^{(*)}$-junctions, that are mostly located at small Ar bubbles have the same configuration for two of the three adjacent RW-AFM domains, whereas the Y$^{(*)}$-junctions, that are not related to the Ar bubbles have a different orientation of all RW-AFM domains, i.e., their parallel spin rows are always radial from the junction center.

## Emergent topological properties of the junctions

The triple- and the hexa-junctions, which form at meeting points of the orientational RW-AFM domains, locally exhibit the 3Q state. This intriguing three-dimensional spin structure has a vanishing net spin moment. However, it is predicted to exhibit topological orbital moments (TOMs)[23,44,57], which are susceptible to applied magnetic fields and lead to the spontaneous topological Hall effect[6,7]. TOMs occur even in the absence of spin-orbit coupling due to electron motion in the emergent magnetic field[57,58]. The TOM can be related to the scalar spin chirality $\chi_{ijk}$ given by $\chi_{ijk} = \mathbf{s}_i \cdot (\mathbf{s}_j \times \mathbf{s}_k)$ for three spins $\mathbf{s}_i$, $\mathbf{s}_j$, $\mathbf{s}_k$ at sites $i,j,k$. For a collinear or coplanar spin structure, the scalar spin chirality vanishes, however, for a non-coplanar spin structure such as the 3Q state, it is finite[48,57]. The TOM at site $i$, $\mathbf{L}_i^{TO}$, is given by[23]

$$\mathbf{L}_i^{TO} = \sum_{(jk)} \kappa_{ijk}^{TO} \chi_{ijk} \boldsymbol{\tau}_{ijk} \tag{1}$$

where $i,j,k$ denote neighboring lattice sites and the sum $(jk)$ is over all pairs of neighboring lattice sites of site $i$. $\kappa_{ijk}^{TO}$ is the topological orbital susceptibility which depends on the electronic structure. The vector $\boldsymbol{\tau}_{ijk} \propto (\mathbf{R}_j - \mathbf{R}_i) \times (\mathbf{R}_k - \mathbf{R}_i)$ is the surface normal of the triangle spanned by the position vectors $\mathbf{R}_i$, $\mathbf{R}_j$, and $\mathbf{R}_k$ of lattice sites $i,j$, and $k$, respectively.

The 3Q state and its topological properties have previously been studied for magnetic monolayers[42-44,48,57]. For a magnetic monolayer $\tau_{ijk}$ is perpendicular to the surface. Two inverted 3Q states with opposite directions of TOMs along the surface normal exist, often referred to as 'all-in' and 'all-out' 3Q state[6,7,44]. For systems beyond a single layer, such as the magnetic bilayer considered here, the 3Q state can occur both with an effective antiferromagnetic (⇄) or ferromagnetic (⇉) coupling between adjacent layers as shown above (cf. Figs. 3a, b and 6d, e). For both types of the bilayer 3Q state we have calculated the TOM for each Mn atom of the non-sheared pseudomorphic Mn double-layer on Ir(111) via DFT. The TOMs of the atoms within a specific layer are constant and point in the same out-of-plane direction. The absolute value of the TOMs is larger for the 3Q$_⇉$ state, however, in that case, the TOMs point in opposite directions in the two layers and nearly cancel for the Mn double-layer (Fig. 6e). This is a consequence of combining an 'all-in' configuration of one layer with an 'all-out' configuration of the other layer. Intriguingly, the antiferromagnetic alignment of the TOMs occurs for a net ferromagnetic coupling of the spin moments.

In contrast, for the energetically favorable 3Q$_⇄$ state (Fig. 6d), with a net antiferromagnetic coupling of the spin moments, the TOMs of the two layers are parallel. This results in a significant total TOM of $0.26\mu_B$/magnetic unit cell. In this configuration, a 3Q state is realized not only within each Mn layer but also between the layers (see tetrahedron in Fig. 6d). Whereas the TOM vanishes for the collinear RW-AFM domains and the coplanar 2Q superposition domain walls, all the triple- and hexa-junctions exhibit the non-coplanar 3Q state and therefore also an associated local TOM. The size of the total TOM is given by the area where the spin texture is non-coplanar. The experiments show that the expected TOM is largest for the highly symmetric hexa-junction, and from the TOM per magnetic unit cell and an

estimated area of 20 nm$^2$ we derive a total TOM of about 20 $\mu_B$ for the hexa-junction shown in Fig. 6a, b.

## Discussion

Our combined experimental and theoretical work shows that the incorporation of local strain can induce the formation of a domain wall network in the antiferromagnetic Mn double-layer on Ir(111). Due to the presence of three symmetry-equivalent orientational antiferromagnetic domains, this network forms triple- and hexa-junctions. The use of SP-STM has enabled us to identify the nature of the different building blocks of the network, i.e., the collinear antiferromagnetic domains, the coplanar domain walls, and the non-coplanar domain wall junctions. DFT calculations have revealed that the magnetic exchange energy is responsible for a significant lateral shift of the top Mn layer with respect to the bottom one. This shearing leads to a structural chirality of triple-junctions, where three domains with different shift directions have to connect, thereby building up strain at the domain walls. The local strain induced by the Ar bubbles is in turn responsible for the preference of specific orientations of the adjacent domains. While also other sources of local strain, such as defect lines (see Fig. S8), can induce domain wall networks, for our system, the Ar bubbles lead to the best connectivity. Our DFT calculations have confirmed that compressive strain at the rim of the Ar bubbles selects one out of the three possible orientational domains. This can be viewed as a reciprocal effect of the magnetism-driven shearing of the magnetic film. Because this magnetism-driven shearing within the antiferromagnet is the origin of the strain-induced network, we expect that the network exists up to the magnetic ordering temperature. While for our system we expect that the ordering temperature is much higher than the roughly 100 K reported for a similar system with only one monolayer[59], we have examined the existence of a domain wall network only up to 80 K (see Fig. S8). Using DFT calculations, we have quantified the size of the topological orbital moments (TOM) arising for the non-collinear magnetic state at the antiferromagnetic domain wall junctions, demonstrating that only antiferromagnetically coupled layers, as is the case here, exhibit significant TOMs.

We anticipate that beyond the fundamental interest in antiferromagnetic domain walls and junctions such networks are interesting candidates for novel computing schemes as they provide several different unique transport properties: (i) the uniaxial collinear magnetic domains are expected to show transport characteristics depending on the angle of the lateral current with respect to the Néel vector, (ii) likely the non-collinear magnetic state in the 120° superposition domain walls strongly interacts with current, (iii) the non-coplanar 3Q junctions give rise to the topological Hall effect. While the latter is expected to cancel for zero-field cooled samples, we expect that the effect can be maximized by field-cooling, which initiates the 3Q state with a TOM parallel to the applied magnetic field. While strong interaction with currents are likely, the network itself is strongly pinned due to the strain-induced hexa-junctions, making it robust against perturbations.

We expect that the generation of such domain wall networks by local strain is not limited to ultra-thin films or other two-dimensional systems. We anticipate that the same mechanism is applicable also in the bulk of antiferromagnets, and also in various material classes. Antiferromagnets in general are highly susceptible to strain: whereas in ferromagnets the coupling of the spin to the lattice occurs only due to spin-orbit interaction, in antiferromagnets the driving force for magneto-elastic deformations is typically the exchange interaction, which is a much larger energy term. In addition to the effects of global strain, as for instance generated by piezoelectric substrates or in multiferroics, there are several possibilities to induce local strain, e.g., by ion implantation, by lithography or other structuring methods. In turn, antiferromagnetic thin films could be used as sensors of local strain at the surface of a sample of interest. Local strain in bulk

antiferromagnets can lead to more complex three-dimensional domain wall networks. We anticipate that our proof-of-principle demonstration will trigger further investigations of the intimate relationship of structure and magnetism. We propose that such domain wall networks can be created in various antiferromagnetic materials, opening the possibility to study their transport properties and response to currents in view of future spintronics applications.

## Methods

### Sample preparation

Samples were prepared in ultra-high vacuum and transferred in-situ to a low-temperature STM. Ir(111) crystals were cleaned by cycles of Ar-ion etching (1.5 keV) and subsequent annealing ($T \approx 1600$ K). To obtain Ar bubbles below the surface, the last annealing step was modified, and a lower annealing temperature of about 1300 K was used[40]. Mn was evaporated from a Knudsen cell held at $T \approx 1050$ K, leading to a deposition rate of about 0.15 atomic layers per minute. The Ir(111) substrate was kept at around 470 K during Mn deposition.

### (Spin-polarized) scanning tunneling microscopy

To obtain information on the magnetic properties of a surface we employ SP-STM, which exploits the spin-polarization of the tunnel current between two magnetic electrodes. The spin-polarized tunnel current scales with the cosine of the angle between tip and sample magnetization directions[41,60,61]. Here we have used a Cr-bulk tip (except for Fig. S5c, for which an Fe-coated W tip was used). Throughout this work we refer to measurements with and without spin-polarized contrast as SP-STM images and STM images, respectively.

The RW-AFM state always appears as stripes, and the magnetic corrugation amplitude depends on the relative tip magnetization direction: when the tip magnetization direction is along the quantization axis of the RW-AFM state, we obtain maximal magnetic signal, whereas an orthogonal orientation results in a vanishing magnetic contrast. The 3Q state is a non-coplanar state, and regardless of the tip magnetization direction this leads to spatial variation of the spin-polarized contribution to the tunnel current. The possible patterns range from a hexagonal p(2 × 2) pattern to a stripe p(2 × 1) pattern[42].

The fact that the domain walls can be seen very prominently in the STM measurements is dominantly due to the spin-averaged contribution to the tunnel current. Different magnetoresistance effects come to mind, for instance the non-collinear magnetoresistance effect, where a signal difference is observed for collinear versus non-collinear magnetic order[62]. However, for this system likely the dominating effect is the local strain at the domain walls. Because the Mn atoms in the RW-AFM domains are located roughly in the bridge sites, and those at the junctions or the domain walls are more closer to threefold hollow sites, this difference in adsorption site is expected to be the main origin for the strong signal difference.

The presented STM images are (plane-fitted) raw data (except when specified otherwise) and were obtained at zero magnetic field (note that Figs. 1a, b, 2c, d, 5, and 6a, b are in the remanent state; however, we do not see any difference between remanent and virgin state). Maps of differential conductance (d$I$/d$U$) were obtained simultaneously to the constant-current images with a lock-in amplifier, and the sample bias voltage was modulated with a peak-to-peak amplitude of about 10% at a frequency on the order of 3−5 kHz.

### Spin models

To illustrate the observed spin structures, we have employed spin dynamics simulations. In the Heisenberg model, the RW-AFM state and the 3Q state are degenerate, both in a monolayer and a double-layer. An energy difference can arise due to higher-order interactions[42,43]. Because an extraction of higher-order interactions from DFT calculations for double-layers has not been possible up to now, the relevant parameters that would serve as input parameters to realistically

describe our system are missing. For this reason, we have used a monolayer setup with generic parameters to illustrate the different building blocks for the network, without the aspiration to perfectly describe the system under study. In particular, for this monolayer setup we also neglected the shift of the top layer with respect to the bottom layer. To set up the system we have used the following Hamiltonian[11]

$$H = -\sum_{i,j} J_{ij}(\mathbf{s}_i \cdot \mathbf{s}_j) - \sum_{i,j} B_{ij}(\mathbf{s}_i \cdot \mathbf{s}_j)^2 \\ - \sum_i K_u(\mathbf{s}_i \cdot \hat{\mathbf{z}})^2 - J_{ASE} \sum_{ij} (\mathbf{s}_i \cdot \mathbf{d}_{ij})(\mathbf{s}_j \cdot \mathbf{d}_{ij})$$

(2)

including nearest neighbor isotropic exchange $J_1$, next-nearest neighbor isotropic exchange $J_2$, …, the biquadratic (or 2-site-4-spin) higher-order term $B$, an unaxial magnetocrystalline anisotropy $K_u$, and the anisotropic symmetric exchange $J_{ASE}$, where $\mathbf{s}_i$ denotes a normalized spin at a lattice site specified by $i$, $\hat{\mathbf{z}}$ is the unit vector perpendicular to the surface and $\mathbf{d}_{ij}$ is the normalized connection vector between the lattice sites $i$ and $j$. The spin dynamics simulations presented in the manuscript were done with Monte Crystal 3.2.0, which can be found on github[63].

The parameters for the different images vary slightly and are: $J_1 = -25, J_2 = -5, J_{ASE} = +0.1, B = +0.25 \pm 0.15, K_u = -0.4$ (all in meV/atom); Fig. 1d is a cut-out of a much larger simulation box and in the case of Figs. 2e and 6c the spins of some edges were fixed throughout the simulation; Fig. 4d is obtained from the spin model in Fig. 2d (rotated by 180°): atoms within the 1Q (RW-AFM) domain atoms are positioned in bridge sites, atoms in the 3Q junction center are positioned in hollow sites, in-between atoms are shifted according to their 1Q components; linear color scale red-yellow by atom shift distance.

### Density functional theory calculations

We employed density functional theory (DFT) using the projected augmented wave (PAW) method[64] as implemented in the VASP code[64–66] to perform both geometry optimizations and calculations of total energies for different magnetic states of Mn double-layers (DLs) on Ir(111). In all calculations, the theoretical in-plane lattice constant of Ir within the generalized gradient approximation (GGA) was used which amounts to 2.75 Å[67] and a large energy cutoff of 400 eV was chosen. Exchange-correlation effects were included by means of the GGA potential with the interpolation developed by Perdew, Burke and Ernzerhof (PBE)[68]. For further computational details of these calculations see the Supplementary Information.

In order to calculate STM images based on DFT (see Fig. 4a, b), we applied the Tersoff-Hamann model[52]. These DFT calculations were performed for the laterally relaxed RW-AFM$_{\rightleftarrows}$ state as well as for the 3Q$_{\rightleftarrows}$ state of the Mn double-layer on Ir(111) using the full-potential linearized augmented planewave (FLAPW) method[69] as implemented in the FLEUR code[70]. Exchange-correlation effects were included in the local density approximation using the parameterization of Vosko, Wilk and Nusair[71]. Besides applying the two-atomic rectangular (four-atomic hexagonal) unit cell of the RW-AFM (3Q) state, an asymmetric slab with nine Ir layers and a Mn DL and 247 $k$-points including the $\bar{\Gamma}$-point (288 $k$-points) in the irreducible part of the BZ were used. The muffin tin radii were set to 2.16 a.u. for Mn in the shifted RW-AFM$_{\rightleftarrows}$ state and to 2.31 a.u. for Ir. Moreover, a large cutoff of $k_{max} = 4.1$ a.u.$^{-1}$ was chosen to ensure convergence with respect to the basis functions. The STM images were calculated for an energy window corresponding to a bias voltage of +100 meV and at a height of 3 Å above the surface.

We have further employed the FLAPW-based FLEUR code to check the results for the shifted RW-AFM$_{\rightleftarrows}$ state obtained via VASP. Applying the above mentioned geometric FLEUR setup, i.e., a two-atomic rectangular unit cell, and the PBE[68] exchange correlation potential, the energy gain of the laterally relaxed structure with respect to the unshifted state is calculated as −64.18 meV/Mn atom, whereas the

corresponding `VASP` value amounts to −57.36 meV/Mn atom (cf. Fig. 3e). Hence, the energy gain obtained via `FLEUR` is on a similar order of magnitude.

## Data availability

The STM data generated in this study have been deposited in the Zenodo database under accession code https://doi.org/10.5281/zenodo.17707861. Other data that support the findings of this study are available from the corresponding author upon reasonable request.

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

## Acknowledgements

This project has received funding from the European Union's Horizon 2020 research and innovation programme under the Marie Skłodowska-Curie grant agreement No 955671 (K.v.B., V.S., A.R.-S.). K.v.B. gratefully acknowledges financial support from the Deutsche Forschungsgemeinschaft (DFG, German Research Foundation) via SPP2137 "Skyrmionics" (project no. 402843438) and project no. 418425860. M.G., So.H., and St.H. gratefully acknowledge financial support from the Deutsche Forschungsgemeinschaft (DFG, German Research Foundation) via SPP2137 "Skyrmionics" (project no. 462602351) and project no. 418425860 and project no. 555842692, and the computing time made available to them on the high-performance computer "Lise" at the NHR center NHR@ZIB. This center is jointly supported by the Federal Ministry of Education and Research and the state governments participating in the NHR (www.nhr-verein.de). We acknowledge financial support from the Open Access Publication Fund of Universität Hamburg.

## Author contributions

V.S. and F.Z. prepared the samples. V.S. performed the measurements. V.S., A.R.-S., and K.v.B. analyzed the data. A.K. performed the spin dynamics simulations and made the spin models. M.G. and So.H. performed the DFT calculations. M.G., So.H., and St.H. analyzed the DFT data. K.v.B., M.G., and St.H. wrote the manuscript. V.S., M.G., A.R.-S., So.H., F.Z., R.W., A.K., St.H., and K.v.B. discussed the results and commented on the manuscript.

## Funding

## Competing interests

The authors declare no competing interests.
