## [Transparent Peer Review file · Nature Communications]

Strain-driven domain wall network with chiral junctions in an antiferromagnet

Corresponding Author: Dr Kirsten von Bergmann

Version 0:

Reviewer comments:

Reviewer #1

(Remarks to the Author)

Note: This review includes comments from another co-reviewer as a part of Nature Communications initiative facilitating training in peer review.

The authors have presented a compelling work demonstrating how induced strain can lead to a complex domain wall network in an anti-ferromagnetic material (AFM). The authors found that the network can host junctions with chiral magnetic properties and potentially interesting topological properties. We are sure the research community will benefit from the publishing of this work. That said, we have listed some comments on scientific content and presentation that we believe will improve the clarity and accessibility of the manuscript to non-specialists.

On scientific content,

1. The paper boldly declares how surface/substrate strain leads to domain wall network (it's in the title and abstract). However, the manuscript is devoid from discussion of how such strain leads to domain wall network. The authors measured the 'diameter' and 'height' of the surface distortion caused by the 'Ar-bubbles located below the surface' (Lines 95-96). Please clarify by additional experiments, analysis, or discussion:

(i) if the Ar-bubble generates a surface curvature (like a hill) (if this is the case, how does this change the inter-atomic spacing of the surface Ir-atom?) or if the Ar-bubble changes the density of state of the surface (which remains flat);

(ii) if there is a threshold/proportional relationship between the diameter and/or height of the surface distortion to the types of the domain wall network motif generated (i.e. Y, Y*, T, T*, or hexa);

(iii) if ALL or SOME junctions (i.e. Y, Y*, T, T*, or hexa) are due to surface distortion induced by Ar-bubbles (please perform analysis (ii) on surfaces with Ar-bubbles and on surfaces without Ar-bubbles and present them in a table, histogram, or something equivalent).

In summary, please provide more atomic-level characterization on the structural origin of the 'strain' that the authors speak of here and how that translates to different types of domain wall network motif.

2. Does the domain wall actually 'map' the surface strain/distortion? If this is true, this could be very exciting as one would be able to deposit these Mn-film on semi-rigid materials (e.g. freestanding semiconducting 2D materials) and could follow in real-time the rippling of materials and hence, modulation of band gaps on various locales of the 2D material. In addition, one could also place such anti-ferromagnetic material on a substrate prepared with Si-towers to generate well-separated junctions, opening new avenues for condensed matter physics experiments. This however can only be revealed with further data analysis (see point 1). We thereby urge the authors to extract these conclusions from their present data and add, as an outlook, exciting future experiments enabled by this discovery.

3. It appears, from Fig. 2e, that the domain walls should meet exactly at some common point. However, Fig 2c shows that the domain walls do not exactly meet in the middle and form as suggested later in the text as a windmill motif. How do the authors reconcile the difference between the experimental data and the DFT results? Also see Figure 4d.

4. It is unclear how the authors relate taking into account the breaking of the three-fold symmetry of the substrate by the RW-AFM state means that the Mn atoms can shift their positions (Lines 170-178). Can the authors clarify this link?

5. For completeness, should the authors not show the calculated spin-averaged STM images based on the Tersoff-Hamann model for the RW-AFM state (unshifted), at least in the SI? Especially, if the authors are using the calculations to help substantiate the claim that the Mn atoms on the top layer is shifted. I do agree, however, that the features purported to be the top-layer Mn atoms (Figure 4c) are not aligned across the domain walls.

6. Given discussion in point 5, would this mean that the Mn double layer at elevated temperatures would host a different structure than the shifted top layer model proposed in this work? Since the origin of the shifted top layer is due to exchange interactions, presumably at higher temperatures, these interactions will play a minimal role (on average)?

On figure presentation,

7. Given the title and abstract of the paper, the manuscript will benefit from a new figure showing the relationship between Ar-bubble-induced surface distortion inducing the domain wall network. This figure should show two (or four) large-scale images of (a) the Mn film without surface/substrate distortion (including before- and after-film growth) and (b) the Mn film with surface/substrate distortion (again, including before- and after-film growth). We are aware that Fig. S1 somewhat accomplishes this, and we suggest to improve Fig. S1 (which is very hard to understand at the moment) and elevate Fig. S1 as the new Fig. 1 of the manuscript.

8. I am unsure as to what is being referred to in Fig. 1c as a hexagonal pattern (mentioned in Lines 106-107). Can the authors be clearer with it by including a guide on the figures? Especially if it appears as a hallmark of the 3Q state in SP-STM.

9. Please add an indication on Fig 1.c on which lines are being referred to in Lines 104-105: "The *lines* visible in the two orientational domains of Fig. 1c,d enclose an angle of 120° , and we find that all of the straight walls in the overview image (Fig. 1a,b) are of this type."

On written presentation, we believe the authors will benefit from restructuring many of the sentences in this work to aid the reader in understanding the actual scientific value of the work rather than leaving the reader to decipher/guess the unclear/ambiguous sentences. There are many instances where a comma/punctuation should be introduced but were mistakenly left out. In addition, there are instances of ambiguous pronouns and ambiguous relative clauses in the manuscript. The authors should consider their text and rewrite to add clarity to their writing. Below are some examples:

1. Missing comma in Line 14: "At the same time, ..."
2. Line 17: "... but [also] to exploit ..."
3. Please rewrite; wrong sentence structure in Line 20: "ranging from collinear magnetic domains, over non-collinear domain walls, to non-coplanar localized domain wall junctions".
4. "for instance regarding" in Line 27 does not make sense. Replace with "such as"
5. Please rewrite. Incoherent sentencing in Lines 28-29: "Next to collinear antiferromagnets also systems with non-collinear magnetic order as in Mn₃Sn [4, 5] or non-coplanar order as in CoTa₃S₆ [6, 7] ...".
6. Missing comma in Line 31: "At the same time, ..."
7. Line 33-34, rewrite to: "size of domains, the properties of antiferromagnetic domain walls are also relevant for spintronics applications"
8. Incoherent sentence. Please rewrite the sentence in lines 46 to 48.
9. Line 54: "In the case of ..."
10. Missing comma in Line 56: "Often, ..."
11. Incoherent sentence structure. Please rewrite the sentence in lines 60 to 63.
12. Missing comma in line 64: "In a spin spiral phase, ..."
13. typo in line 68: "A large-scale patterning ..."
14. Rephrase line 70: "... which manifest for instance in ... " to "... which manifest in, for instance, ..."
15. . Please rewrite; wrong sentence structure in Line 77: "ranging from collinear magnetic domains, over non-collinear domain walls, to non-coplanar localized domain wall junctions".
16. Missing comma in line 92: "In this system, ..."
17. Missing comma in line 98: "Here, ..."
18. Line 121: "adapt" instead of "adopt"?
19. Line 144: "Note, that within all DFT, ..."
20. Line 243: "After the analysis of the magnetic state in the triple-junctions we turn, ..."

Reviewer #2

(Remarks to the Author)

Reviewer #3

(Remarks to the Author)

The authors studied the spin configurations of double-layer Mn using SP-STM. According to the symmetry of the material, there are six different collinear ground states. Their main findings are as follows. First, the collinear ground states involve the lateral shift of $\sim 1 \text{ \AA}$ of Mn atoms in one of the two layers, verified by the DFT calculations. Second, two ground states meet along a domain wall, across which the spin configurations change in a co-planar fashion. The displacement vector of the moved Mn atoms changes across the domain wall too. Thirdly, three ground states meet at one junction point, around which the spin structures are non-coplanar. In this case, the three displacement vectors of the involved three ground states rotate around the junction point, showing the specific structural chirality. Overall, they found that domain walls can form a network with multiple junctions with characteristic structural chirality, which they called 'strain-driven domain wall network with chiral junctions.' While it is well-known that domain walls can occur naturally in various magnetic materials, the question of whether domain walls can form a network in magnetic systems has been studied little despite its fundamental interest as well as technological relevance. The current manuscript reports clear evidence of domain-wall networks with rich topological characteristics in antiferromagnets, which will advance our fundamental understanding of magnetism. It has also potential utility in spintronics as a platform over which spin information can flow. For this reason, in principle, I can recommend the publication of the manuscript in Nature Communications, provided that the following comments can be addressed.

1) While the authors focused on the chirality of the Mn-atom displacement vectors at the junctions, the spin configurations around the triple junctions can be considered as half-skyrmions or merons of the Neel-order parameter, which can be characterized by the topological winding number (see, e.g., [Yu et al., "Transformation between meron and skyrmion topological spin textures in a chiral magnet," Nature 564, 95 (2018); Jane et al., "Antiferromagnetic half-skyrmions and bimerons at room temperature," Nature 590, 74 (2021)]). In the current manuscript, the triple junctions are characterized only by the geometric Y/T type and by the patterns of the structural displacement vectors. It would be instructive to characterize triple junctions by their winding numbers, which are rooted in spin degrees of freedom.

2) The reason why Y and T types always occur together can be understood by the vortex-antivortex interaction in the XY systems. From Fig. 5, around the Y and Y* junction, the displacement vectors (blue arrows) rotate counterclockwise as we encircle the junction counterclockwise. On the contrary, around the T and T* junctions, the displacement vectors rotate clockwise as we encircle the junction counterclockwise. This indicates that Y/Y* junctions are vortices, whereas the T/T* junctions are antivortices in the analog of the XY model. The observation that Y types and T types are generated together can be explained easily by the well-known pairwise generation of vortices and antivortices in the XY model. I suggest that the authors incorporate this analogy in the manuscript to facilitate readers' understanding.

3) The authors calculated the topological orbital moment (TOM) for one type of hexa-junctions. However, TOM is expected to be finite for all types of triple junctions as well as all types of hexa-junctions. Furthermore, TOM should depend on the type of junctions, such as Y, Y*, T, T* type for triple junctions. Can the authors show the TOM calculation results for all different types of triple junctions and hexa-junctions? This will enable distinguishing different types of junctions.

4) Please provide the 'color scheme' for spin configurations such as Fig. 2e and Fig. 6c.

Reviewer #4

(Remarks to the Author)

The manuscript presents a high-quality study that combines spin-polarized scanning tunneling microscopy and first-principles calculations to investigate a strain-induced domain wall network in a Mn double-layer on Ir(111). The authors convincingly identify a range of magnetic building blocks, including collinear antiferromagnetic domains, non-collinear domain walls, and non-coplanar 3Q-type triple junctions. The discovery of chiral junctions exhibiting topological orbital magnetization is particularly compelling and provides valuable insight into the physics of emergent topological spin textures in antiferromagnets.

I regard this work as scientifically rigorous and potentially impactful, especially in the areas of topological magnetism and emergent electronic phenomena. The identification of chiral triple junctions hosting non-coplanar 3Q spin textures and associated topological orbital magnetization represents a novel and significant result that will be of strong interest to the fundamental physics community.

However, I do not believe the manuscript fully aligns with the scope of this journal. Its emphasis is primarily on fundamental physical mechanisms, and its appeal may be more limited in terms of broad, interdisciplinary readership. For these reasons, I recommend the authors consider submitting their work to Communications Physics, where the journal's focus on foundational physical insights and emergent topological effects is likely to be a better match.

General comments:

1. The study is performed at ultra-low temperature ($T=4 \text{ K}$). While this enables high-resolution imaging, it raises questions about the relevance and stability of the observed domain wall network and 3Q states at technologically relevant or even intermediate temperatures. The lack of any temperature-dependent study limits the broader applicability of the findings.
2. Although the manuscript touches on potential transport phenomena (topological Hall effects), no experimental verification

of these properties is provided. This weakens the impact for journals with an application-oriented or broader audience.

3. The introduction, while thorough, can benefit from a more focused motivation paragraph early on to guide readers unfamiliar with the domain.

4. Some sections could be shortened or streamlined, especially those reviewing prior work, this could help draw focus to the new contributions.

5. The energetic contradiction between DFT predicting the 3Q state as lowest-energy and experimental observation of RW-AFM domains is well-resolved by considering structural relaxation. However, the paper could do more to explain this transition physically, especially for non-specialists.

6. While the paper ends with strong conclusions, it would benefit from a brief discussion of experimental limitations, for example the field, temperature, thickness dependence.

Minor comments:

1. Figure references sometimes appear before the figure is described in sufficient detail.

2. Add clear units to colorbars and vertical axes in DFT energy plots.

Version 2:

Reviewer comments:

Reviewer #1

(Remarks to the Author)

Note: This review includes comments from another co-reviewer as a part of Nature Communications initiative facilitating training in peer review.

We thank the authors in implementing some of our suggestions, which we think have clarified the main findings of the paper. We congratulate the authors for this discovery and we hereby endorse the acceptance and publication of the manuscript in Nature Communications.

Reviewer #2

(Remarks to the Author)

Reviewer #3

(Remarks to the Author)

The authors have responded well to the comments of the reviewers, including myself, leading to the recommendation of the publication of the manuscript in Nature Communications.

Reviewer #4

(Remarks to the Author)

After reviewing the author's rebuttal letter and the revised manuscript, I would like to reiterate and clarify my concerns:

My initial comment addressed the lack of finite temperature measurements, as the entire study is conducted at 4K. The authors did not provide a clear explanation for why this temperature was chosen, nor did they discuss the potential behavior of the strain-induced domain wall network at higher temperatures. I am particularly interested in whether this domain wall network persists at larger temperatures. While it is valid to conduct a fundamental study at very low temperatures, the authors must clarify that the scope of this work is limited to 4K and does not necessarily have direct applications at higher temperatures. Their response to this comment was unsatisfactory, as they failed to explain the rationale for using 4K or provide any discussion of how the results might change at different temperatures. A clearer justification is needed, especially if the study aims to contribute to applications in a broader temperature range.

Additionally, the authors did not adequately address comments 4 and 6 from my previous review. Their replies to these points were either insufficient or absent. These comments were central to the clarity and depth of the manuscript, and a more thorough response is required to address the underlying issues.

Version 3:

Reviewer comments:

Reviewer #4

(Remarks to the Author)

After this last review the authors have addressed my comments and clarified the critical point about temperature measurements. The manuscript can be considered for publication in Nature Communications.

REVIEWER COMMENTS

Reviewer #1 (Remarks to the Author):

Note: This review includes comments from another co-reviewer as a part of Nature Communications initiative facilitating training in peer review.

The authors have presented a compelling work demonstrating how induced strain can lead to a complex domain wall network in an anti-ferromagnetic material (AFM). The authors found that the network can host junctions with chiral magnetic properties and potentially interesting topological properties. We are sure the research community will benefit from the publishing of this work. That said, we have listed some comments on scientific content and presentation that we believe will improve the clarity and accessibility of the manuscript to non-specialists.

We would like to thank the reviewers for the positive evaluation of our manuscript. We will respond to the listed suggestions on how to improve the clarity and accessibility point-by-point.

On scientific content,

1. The paper boldly declares how surface/substrate strain leads to domain wall network (it's in the title and abstract). However, the manuscript is devoid from discussion of how such strain leads to domain wall network. The authors measured the 'diameter' and 'height' of the surface distortion caused by the 'Ar-bubbles located below the surface' (Lines 95-96). Please clarify by additional experiments, analysis, or discussion:

(i) if the Ar-bubble generates a surface curvature (like a hill) (if this is the case, how does this change the inter-atomic spacing of the surface Ir-atom?) or if the Ar-bubble changes the density of state of the surface (which remains flat);

Based on previous works, e.g. by Gsell et al. (Ref. [40] in the current version of the manuscript) the Ar bubbles are expected to lead to protrusions (hills) in the topography. We are aware that this interpretation may not be valid for all single crystal substrates, as there are other references that suggest a flat surface. As the STM technique only measures apparent height, which may depend on the applied sample bias voltage and the work function, one cannot rigorously prove one or the other scenario with constant-current measurements. In contrast to a previous study of fcc-Al(111) with Ar bubbles, where a voltage dependent variation of the apparent height at the Ar bubbles was found [Schmid et al., Phys. Rev. Lett. 76, 2298 (1996)], we always see the Ar bubbles in Ir(111) as protrusions, regardless of the bias voltage. The previous observation on Al(111) can be explained by quantum well states, which is specific for that system and absent for Ir(111). Furthermore, the Ar bubbles appear as protrusions not only within the pristine Ir(111) (as shown in Fig. S1b), but also when they are below the Mn double-layer grown on Ir(111) (as shown for instance in Fig. S1c), or even below a Mn triple-layer (not shown). Therefore we conclude that in our case the Ar bubbles lead to a protrusion/hill-like structure at the surface. We have added this information to the caption of Fig. S1. While the estimated change in inter-atomic spacing for a protrusion of 50 pm height and 7 nm diameter is very small, it is enough to break the symmetry. The scenario of a hill-like Ar bubble is illustrated in the supplementary Fig. S5a as a side view. To further illustrate the meaning of the presented DFT calculations of the strained film we have added another top view sketch in Fig. S5b.

(ii) if there is a threshold/proportional relationship between the diameter and/or height of the surface distortion to the types of the domain wall network motif generated (i.e. Y, Y*, T, T*, or hexa);

As the Y- and Y*-junctions are degenerate and also the T- and T*- are degenerate we expect no difference for those pairs. On a qualitative note we can say that larger Ar bubbles typically generate hexa-junctions, which can be viewed as a pair of T/T*-junctions, whereas medium bubbles often only host one T- or T*-junction. We have added this aspect to the manuscript. A quantitative analysis would require a much larger statistics and is beyond the scope of this work. The Ar bubbles do not induce or pin Y- and Y*-junctions, which we qualitatively explain in the revised and extended section 'Origin of the Network'.

(iii) if ALL or SOME junctions (i.e. Y, Y*, T, T*, or hexa) are due to surface distortion induced by Ar-bubbles (please perform analysis (ii) on surfaces with Ar-bubbles and on surfaces without Ar-bubbles and present them in a table, histogram, or something equivalent).

All hexa-junctions and T- and T*-junctions are located at Ar bubbles. However, some shallow Ar bubbles do not have a domain wall junction located on them. The Y- and Y*-junctions are not related to the Ar bubbles, but due to the requirements of the connectivity of the network they must form to connect the hexa-, T-, and T*-junctions. We see experimentally (Fig. 1 and Fig. 6) and demonstrate by DFT (Fig. S5) that the compressive strain around the Ar bubbles favors the orientational RW-AFM domain with tangential stripes (see revised Fig. S5). Note that in Y- and Y*-junctions the stripes run in the radial directions, therefore they are not expected to be induced by the Ar bubbles, as also observed experimentally.

In summary, please provide more atomic-level characterization on the structural origin of the 'strain' that the authors speak of here and how that translates to different types of domain wall network motif.

We have extended our discussion on the role of the Ar-bubbles for the generation of the domain wall network and more explicitly mentioned the results shown in Fig. S5 also in the main text (see revised and extended section 'Origin of the Network'). To clarify the relation between the Ar bubble and the selection of a specific orientational domain we have added a top view sketch in Fig. S5b. The sketch illustrates that the strain around the bubble leads to the characteristic alignment of the domains with respect to the junction core.

2. Does the domain wall actually 'map' the surface strain/distortion? If this is true, this could be very exciting as one would be able to deposit these Mn-film on semi-rigid materials (e.g. freestanding semiconducting 2D materials) and could follow in real-time the rippling of materials and hence, modulation of band gaps on various locales of the 2D material. In addition, one could also place such anti-ferromagnetic material on a substrate prepared with Si-towers to generate well-separated junctions, opening new avenues for condensed matter physics experiments. This however can only be revealed with further data analysis (see point 1). We thereby urge the authors to extract these conclusions from their present data and add, as an outlook, exciting future experiments enabled by this discovery.

We have shown that local strain can induce the formation of particular orientational domains in antiferromagnets. The mechanism for this is understood to be the exchange-interaction-induced shift between layers, which uniquely couples to the strain. In our case the magnetic domain walls are thus located at positions in the film where the shear changes its direction. Based on our findings for our model systems, we anticipate that all phenomena suggested by the reviewer are possible. However,

the magnetic state of the film is governed by many aspects, e.g. symmetry, atomic distances, and hybridization at interfaces, so only experiments will be able to verify if this works for a particular system, or if maybe alternative antiferromagnetic materials might better reflect the strain in a given substrate. We expect that our findings will initiate further research in this area, along the lines of the reviewer's suggestion and added a sentence to the discussion section:

'... In turn antiferromagnetic thin films could be used as sensors of local strain at the surface of a sample of interest. We anticipate that our proof-of-principle demonstration will trigger further investigations of the intimate relationship of structure and magnetism. ...'

3. It appears, from Fig. 2e, that the domain walls should meet exactly at some common point. However, Fig 2c shows that the domain walls do not exactly meet in the middle and form as suggested later in the text as a windmill motif. How do the authors reconcile the difference between the experimental data and the DFT results? Also see Figure 4d.

As the system is quite complex we try to guide the reader through the different aspects one by one. Figure 2e,f are spin dynamics simulations of a symmetric unshifted monolayer, and serves to help the reader understand the spin texture at triple-junctions. Indeed, this is the spin texture that is expected for any system, shifted or not shifted, monolayer or bilayer, in the area where the three orientational domains of the RW-AFM state meet. The reviewer is correct that already in Fig. 2c,d one can see the asymmetry of the double-layer system under study here. However, we decided to guide the reader through the manuscript discussing one aspect at a time, leaving the discussion about the magnetic-exchange-driven shift to Fig. 4, after the origin has been revealed by DFT calculations in Fig. 3. We hope the reviewer will understand our decision to leave this unchanged. However, we have added one sentence in the methods section to prevent any misunderstandings. The additional sentence in the 'Methods: Spin models' section is marked in **bold** text below:

'...we have used a monolayer setup with generic parameters to illustrate the different building blocks for the network, without the aspiration to perfectly describe the system under study. **In particular, for this monolayer setup we also neglected the shift of the top layer with respect to the bottom layer.** To set up the system we have used the following Hamiltonian... '

4. It is unclear how the authors relate taking into account the breaking of the three-fold symmetry of the substrate by the RW-AFM state means that the Mn atoms can shift their positions (Lines 170-178). Can the authors clarify this link?

The expected structural symmetry of the paramagnetic system is C₃ (three-fold rotational axis perpendicular to the surface) with Mn atoms residing in hollow sites. However, the symmetry of the antiferromagnetic state is lower, and in the combined system of antiferromagnetic rows with Mn atoms residing in hollow sites only one mirror plane perpendicular to the stripes remains, see Fig. 3c, where the mirror plane includes the horizontal axis. Any further atom relaxations that maintain this symmetry are now allowed, including a shift of the top layer parallel to the remaining mirror plane.

We have added parts of this discussion in the context of Fig. 3, and now the text reads (new text indicated in **bold**):

'However, we have not yet taken into account that the RW-AFM state breaks the three-fold symmetry of the substrate. **In the system displayed in Fig. 3c only one mirror plane perpendicular to the dotted line remains. As long as this symmetry is maintained** the Mn atoms may in principle shift out of the hollow sites, **which may lead to a** lower total energy of this magnetic state [48].'

5. For completeness, should the authors not show the calculated spin-averaged STM images based

on the Tersoff-Hamann model for the RW-AFM state (unshifted), at least in the SI? Especially, if the authors are using the calculations to help substantiate the claim that the Mn atoms on the top layer is shifted. I do agree, however, that the features purported to be the top-layer Mn atoms (Figure 4c) are not aligned across the domain walls.

We thank the reviewer for pointing out that this data is missing. In the revised version of the Supplementary Information we have added Fig. S6 which shows the calculated spin-averaged STM image for the RW-AFM state for the unshifted Mn double layer.

6. Given discussion in point 5, would this mean that the Mn double layer at elevated temperatures would host a different structure than the shifted top layer model proposed in this work? Since the origin of the shifted top layer is due to exchange interactions, presumably at higher temperatures, these interactions will play a minimal role (on average)?

Correct, we expect that in the paramagnetic state the Mn atoms reside in hollow sites.

On figure presentation,

7. Given the title and abstract of the paper, the manuscript will benefit from a new figure showing the relationship between Ar-bubble-induced surface distortion inducing the domain wall network. This figure should show two (or four) large-scale images of (a) the Mn film without surface/substrate distortion (including before- and after-film growth) and (b) the Mn film with surface/substrate distortion (again, including before- and after-film growth). We are aware that Fig. S1 somewhat accomplishes this, and we suggest to improve Fig. S1 (which is very hard to understand at the moment) and elevate Fig. S1 as the new Fig. 1 of the manuscript.

We prefer to keep the structure of the manuscript as is. Our main findings comprise strain-induced domain wall generation with specific junction types and connectivity, and non-coplanar triple- and hexa-junctions with topological orbital moments. We discuss the mechanism of strain-induced domain selection and alignment with respect to the junctions in the context of Fig. S5. We consider the surface science related details of the distribution of Ar bubble size not particularly important for our main conclusions.

8. I am unsure as to what is being referred to in Fig. 1c as a hexagonal pattern (mentioned in Lines 106-107). Can the authors be clearer with is by including a guide on the figures? Especially if it appears as a hallmark of the 3Q state in SP-STM.

We have rephrased the indicated sentence to clarify this (**bold** indicates the revised text):

'... **Towards** the center of this domain wall **the lines transition into a dotted pattern which locally has hexagonal symmetry**. Thus we conclude that they are **2Q**-superposition domain walls as previously observed for the RW-AFM state in the Mn monolayer on Re(0001) [11]. ...'

9. Please add an indication on Fig 1.c on which lines are being referred to in Lines 104-105: "The *lines* visible in the two orientational domains of Fig. 1c,d enclose an angle of 120°, and we find that all of the straight walls in the overview image (Fig. 1a,b) are of this type."

We have rephrased the indicated sentence to clarify this (**bold** indicates the revised text):

'... The RW-AFM state is identified in **SP-STM images** by its characteristic appearance of alternating bright and dark **lines, which correspond to atomic rows of opposite spin directions** [41, 42], see also **the to-scale** illustration of the magnetic state in Fig. 1d. The lines visible in the two orientational...'

On written presentation, we believe the authors will benefit from restructuring many of the sentences in this work to aid the reader in understanding the actual scientific value of the work rather than leaving the reader to decipher/guess the unclear/ambiguous sentences. There are many instances where a comma/punctuation should be introduced but were mistakenly left out. In addition, there are instances of ambiguous pronouns and ambiguous relative clauses in the manuscript. The authors should consider their text and rewrite to add clarity to their writing. Below are some examples:

We thank the reviewer for the thorough assessment of our manuscript and have carefully considered the suggestions below.

1. Missing comma in Line 14: "At the same time, ..."
2. Line 17: "... but [also] to exploit ..."
3. Please rewrite; wrong sentence structure in Line 20: "ranging from collinear magnetic domains, over non-collinear domain walls, to non-coplanar localized domain wall junctions".
4. "for instance regarding" in Line 27 does not make sense. Replace with "such as"
5. Please rewrite. Incoherent sentencing in Lines 28-29: "Next to collinear antiferromagnets also systems with non-collinear magnetic order as in Mn₃Sn [4, 5] or non-coplanar order as in CoTa₃S₆ [6, 7] ...".
6. Missing comma in Line 31: "At the same time, ..."
7. Line 33-34, rewrite to: "size of domains, the properties of antiferromagnetic domain walls are also relevant for spintronics applications"
8. Incoherent sentence. Please rewrite the sentence in lines 46 to 48.
9. Line 54: "In the case of ..."
10. Missing comma in Line 56: "Often, ..."
11. Incoherent sentence structure. Please rewrite the sentence in lines 60 to 63.
12. Missing comma in line 64: "In a spin spiral phase, ..."
13. typo in line 68: "A large-scale patterning ..."
14. Rephrase line 70: "... which manifest for instance in ... " to "... which manifest in, for instance, ..."
15. . Please rewrite; wrong sentence structure in Line 77: "ranging from collinear magnetic domains, over non-collinear domain walls, to non-coplanar localized domain wall junctions".
16. Missing comma in line 92: "In this system, ..."
17. Missing comma in line 98: "Here, ..."
18. Line 121: "adapt" instead of "adopt"?
19. Line 144: "Note, that within all DFT, ..."
20. Line 243: "After the analysis of the magnetic state in the triple-junctions we turn, ..."

Reviewer #2 (Remarks to the Author):

Reviewer #3 (Remarks to the Author):

The authors studied the spin configurations of double-layer Mn using SP-STM. According to the symmetry of the material, there are six different collinear ground states. Their main findings are as follows. First, the collinear ground states involve the lateral shift of $\sim 1 \text{ \AA}$ of Mn atoms in one of the two layers, verified by the DFT calculations. Second, two ground states meet along a domain wall, across which the spin configurations change in a co-planar fashion. The displacement vector of the moved Mn atoms changes across the domain wall too. Thirdly, three ground states meet at one junction point, around which the spin structures are non-coplanar. In this case, the three displacement vectors of the involved three ground states rotate around the junction point, showing the specific structural chirality. Overall, they found that domain walls can form a network with multiple junctions with characteristic structural chirality, which they called 'strain-driven domain wall network with chiral junctions.' While it is well-known that domain walls can occur naturally in various magnetic materials, the question of whether domain walls can form a network in magnetic systems has been studied little despite its fundamental interest as well as technological relevance. The current manuscript reports clear evidence of domain-wall networks with rich topological characteristics in antiferromagnets, which will advance our fundamental understanding of magnetism. It has also potential utility in spintronics as a platform over which spin information can flow. For this reason, in principle, I can recommend the publication of the manuscript in Nature Communications, provided that the following comments can be addressed.

We thank the referee for carefully reviewing our work, the positive evaluation of our manuscript, and for pointing out its relevance for researchers working in magnetism and spintronics. Based on the reviewer's valuable comments and suggestions we have further improved our manuscript.

1) While the authors focused on the chirality of the Mn-atom displacement vectors at the junctions, the spin configurations around the triple junctions can be considered as half-skyrmions or merons of the Neel-order parameter, which can be characterized by the topological winding number (see, e.g., [Yu et al., "Transformation between meron and skyrmion topological spin textures in a chiral magnet," Nature 564, 95 (2018); Jane et al., "Antiferromagnetic half-skyrmions and bimerons at room temperature," Nature 590, 74 (2021)]). In the current manuscript, the triple junctions are characterized only by the geometric Y/T type and by the patterns of the structural displacement vectors. It would be instructive to characterize triple junctions by their winding numbers, which are rooted in spin degrees of freedom.

We are reluctant to use the spin degree of freedom for the characterization of the junction as the spin texture changes from 1Q (RW-AFM) in the domains to 3Q at the junction, making a coherent definition difficult. In particular in our opinion this also prevents an analogy to merons because in our case the core exhibits a different type of spin structure compared to the outside.

Instead, to connect to the next point of the reviewer, in the new version we now use the different helicity of the shift directions for the different triple junctions to guide the reader through the aspect of structural chirality (see last paragraph of section 'Characterization and Chirality of the Junctions').

2) The reason why Y and T types always occur together can be understood by the vortex-antivortex interaction in the XY systems. From Fig. 5, around the Y and Y* junction, the displacement vectors (blue arrows) rotate counterclockwise as we encircle the junction counterclockwise. On the contrary, around the T and T* junctions, the displacement vectors rotate clockwise as we encircle the junction counterclockwise. This indicates that Y/Y* junctions are vortices, whereas the T/T* junctions are

antivortices in the analog of the XY model. The observation that Y types and T types are generated together can be explained easily by the well-known pairwise generation of vortices and antivortices in the XY model. I suggest that the authors incorporate this analogy in the manuscript to facilitate readers' understanding.

As discussed in our answer to the previous point in the revised version of the manuscript we use the helicity of the shift directions to discuss the structural chirality. Accordingly, we explain the connectivity based on the helicity (see last paragraph of section 'Characterization and Chirality of the Junctions').

3) The authors calculated the topological orbital moment (TOM) for one type of hexa-junctions. However, TOM is expected to be finite for all types of triple junctions as well as all types of hexa-junctions. Furthermore, TOM should depend on the type of junctions, such as Y, Y*, T, T* type for triple junctions. Can the authors show the TOM calculation results for all different types of triple junctions and hexa-junctions? This will enable distinguishing different types of junctions.

DFT calculations demonstrate that the TOM per magnetic unit cell of the 3Q state is about $0.26 \mu_B$. The total topological charge for each junction is then given by the number of 3Q unit cells. Experimentally we find that the area of the hexagonal pattern indicative of the 3Q state is in general largest for the hexa-junctions, whereas the 3Q area in the triple-junctions is smaller. This suggests that the hexa-junctions can give rise to the largest total TOM. According to our spin dynamics simulations this is expected for the symmetric hexa-junction type as shown in Fig. 6c, whereas in other hexa-junctions as displayed in Fig. S3e,f,g the TOM may cancel.

We have added some of this information in the Discussion section (bold indicates the revised text):

'... all the triple- and hexa-junctions exhibit the non-coplanar 3Q state and therefore also an associated local TOM. **The size of the total TOM is given by the area where the spin texture is non-coplanar. The experiments show that the expected TOM is largest for the highly symmetric hexa-junction, and from the TOM per magnetic unit cell and an estimated area of 20 nm^2 we derive a total TOM of about $20 \mu_B$ for the hexa-junction shown in Fig. 6a,b.**'

4) Please provide the 'color scheme' for spin configurations such as Fig. 2e and Fig. 6c.

We thank the reviewer for pointing out that this information is missing and we have added it to the respective captions.

Reviewer #4 (Remarks to the Author):

The manuscript presents a high-quality study that combines spin-polarized scanning tunneling microscopy and first-principles calculations to investigate a strain-induced domain wall network in a Mn double-layer on Ir(111). The authors convincingly identify a range of magnetic building blocks, including collinear antiferromagnetic domains, non-collinear domain walls, and non-coplanar 3Q-type triple junctions. The discovery of chiral junctions exhibiting topological orbital magnetization is particularly compelling and provides valuable insight into the physics of emergent topological spin textures in antiferromagnets.

I regard this work as scientifically rigorous and potentially impactful, especially in the areas of topological magnetism and emergent electronic phenomena. The identification of chiral triple junctions hosting non-coplanar 3Q spin textures and associated topological orbital magnetization represents a novel and significant result that will be of strong interest to the fundamental physics community.

However, I do not believe the manuscript fully aligns with the scope of this journal. Its emphasis is primarily on fundamental physical mechanisms, and its appeal may be more limited in terms of broad, interdisciplinary readership. For these reasons, I recommend the authors consider submitting their work to Communications Physics, where the journal's focus on foundational physical insights and emergent topological effects is likely to be a better match.

We thank the reviewer for evaluating our manuscript and for the positive comments, in particular the statement that our results are novel, significant, and interesting for the fundamental physics community. On the contrary with the reviewer's judgement we believe that this perfectly aligns with the scope of Nature Communications, which has published many papers that provide deep insight into fundamental physics.

General comments:

1. The study is performed at ultra-low temperature ($T=4$ K). While this enables high-resolution imaging, it raises questions about the relevance and stability of the observed domain wall network and 3Q states at technologically relevant or even intermediate temperatures. The lack of any temperature-dependent study limits the broader applicability of the findings.

This study serves as a proof of principle study on a model-type system which allows atomic-scale imaging of all constituents of the domain wall network using spin-polarized STM. We anticipate that strain-induced generation of antiferromagnetic domain walls may be applicable to several other sample systems.

2. Although the manuscript touches on potential transport phenomena (topological Hall effects), no experimental verification of these properties is provided. This weakens the impact for journals with an application-oriented or broader audience.

We do believe that the broader audience of Nature Communications has a strong interest in fundamental physical phenomena. We agree that complementary methods, in particular transport measurements, would be very exciting, however, this is beyond the scope of our work. We hope that our detailed study will trigger such experiments on other appropriate systems.

3. The introduction, while thorough, can benefit from a more focused motivation paragraph early on to guide readers unfamiliar with the domain.

In the revised version we have added a paragraph on the properties and formation of superposition states to guide readers unfamiliar with this concept.

4. Some sections could be shortened or streamlined, especially those reviewing prior work, this could help draw focus to the new contributions.

We have considered this comment, however, we consider the prior work reviewed in our introduction relevant for the context of our work, so we were reluctant to shorten it.

5. The energetic contradiction between DFT predicting the 3Q state as lowest-energy and experimental observation of RW-AFM domains is well-resolved by considering structural relaxation. However, the paper could do more to explain this transition physically, especially for non-specialists.

In our manuscript, we have explained the shift of the top Mn layer by the antiferromagnetic exchange interaction between nearest neighbor Mn atoms of top and bottom layer. We believe that this is at a level that can be understood also by non-specialists. However, we have added a reference to a manuscript recently submitted to PRB in which we present the DFT calculations of spin spirals for the pseudomorphic Mn double layer on Ir(111) (B. Beyer et al., arxiv:2506:05091). In that work, we have determined the inter- and intralayer exchange constants explicitly.

6. While the paper ends with strong conclusions, it would benefit from a brief discussion of experimental limitations, for example the field, temperature, thickness dependence.

Our work serves as a proof-of-principle on a model-type system. Future studies will show how this can be transferred to other systems of interest. We have added this to the discussion section.

Minor comments:

1. Figure references sometimes appear before the figure is described in sufficient detail.

We are unsure what the reviewer is referring to.

2. Add clear units to colorbars and vertical axes in DFT energy plots.

The color scales of the topographic images and the current images are provided in the respective captions (we have added that of Fig. 4c, which was missing). We are unsure which vertical axes of the DFT energy plots are unclear.

We thank the reviewer for the general as well as the minor comments and have considered all of them carefully.

REVIEWER COMMENTS

Reviewer #1 (Remarks to the Author):

Note: This review includes comments from another co-reviewer as a part of Nature Communications initiative facilitating training in peer review.

The authors have presented a compelling work demonstrating how induced strain can lead to a complex domain wall network in an anti-ferromagnetic material (AFM). The authors found that the network can host junctions with chiral magnetic properties and potentially interesting topological properties. We are sure the research community will benefit from the publishing of this work. That said, we have listed some comments on scientific content and presentation that we believe will improve the clarity and accessibility of the manuscript to non-specialists.

We would like to thank the reviewers for the positive evaluation of our manuscript. We will respond to the listed suggestions on how to improve the clarity and accessibility point-by-point.

On scientific content,

1. The paper boldly declares how surface/substrate strain leads to domain wall network (it's in the title and abstract). However, the manuscript is devoid from discussion of how such strain leads to domain wall network. The authors measured the 'diameter' and 'height' of the surface distortion caused by the 'Ar-bubbles located below the surface' (Lines 95-96). Please clarify by additional experiments, analysis, or discussion:

(i) if the Ar-bubble generates a surface curvature (like a hill) (if this is the case, how does this change the inter-atomic spacing of the surface Ir-atom?) or if the Ar-bubble changes the density of state of the surface (which remains flat);

Based on previous works, e.g. by Gsell et al. (Ref. [40] in the current version of the manuscript) the Ar bubbles are expected to lead to protrusions (hills) in the topography. We are aware that this interpretation may not be valid for all single crystal substrates, as there are other references that suggest a flat surface. As the STM technique only measures apparent height, which may depend on the applied sample bias voltage and the work function, one cannot rigorously prove one or the other scenario with constant-current measurements. In contrast to a previous study of fcc-Al(111) with Ar bubbles, where a voltage dependent variation of the apparent height at the Ar bubbles was found [Schmid et al., Phys. Rev. Lett. 76, 2298 (1996)], we always see the Ar bubbles in Ir(111) as protrusions, regardless of the bias voltage. The previous observation on Al(111) can be explained by quantum well states, which is specific for that system and absent for Ir(111). Furthermore, the Ar bubbles appear as protrusions not only within the pristine Ir(111) (as shown in Fig. S1b), but also when they are below the Mn double-layer grown on Ir(111) (as shown for instance in Fig. S1c), or even below a Mn triple-layer (not shown). Therefore we conclude that in our case the Ar bubbles lead to a protrusion/hill-like structure at the surface. We have added this information to the caption of Fig. S1, and line profiles of different Ar bubbles can be found in the new Supplementary Fig. S6 of the revised supplementary information. While the estimated change in inter-atomic spacing for a protrusion of 50 pm height and 7 nm diameter is very small, it is enough to break the symmetry. The scenario of a hill-like Ar bubble is illustrated in the supplementary Fig. S7a as a side view. To further illustrate the meaning of the presented DFT calculations of the strained film we have added another top view sketch in Fig. S7b.

(ii) if there is a threshold/proportional relationship between the diameter and/or height of the surface distortion to the types of the domain wall network motif generated (i.e. Y, Y*, T, T*, or hexa);

As the Y- and Y*-junctions are degenerate and also the T- and T*- are degenerate we expect no difference for those pairs. Our new analysis presented in Supplementary Fig. S6 shows that larger Ar bubbles typically generate hexa-junctions, which can be viewed as a pair of T/T*-junctions, whereas medium bubbles often only host one T- or T*-junction. We have also added this aspect to the main text (see revised and extended section 'Origin of the Network'). The Ar bubbles do not induce or pin Y- and Y*-junctions, which we qualitatively explain in the revised and extended section 'Origin of the Network'.

(iii) if ALL or SOME junctions (i.e. Y, Y*, T, T*, or hexa) are due to surface distortion induced by Ar-bubbles (please perform analysis (ii) on surfaces with Ar-bubbles and on surfaces without Ar-bubbles and present them in a table, histogram, or something equivalent).

All hexa-junctions and nearly all T- and T*-junctions are located at Ar bubbles. However, some shallow Ar bubbles do not have a domain wall junction located on them, but sometimes a wall passing in the vicinity. The Y- and Y*-junctions are not related to the Ar bubbles, but due to the requirements of the connectivity of the network they must form to connect the hexa-, T-, and T*-junctions. We see experimentally (Fig. 1 and Fig. 6) and demonstrate by DFT (Supplementary Fig. S7) that the compressive strain around the Ar bubbles favors the orientational RW-AFM domain with tangential stripes (see revised Supplementary Fig. S7). Note that in Y- and Y*-junctions the stripes run in the radial directions, therefore they are not expected to be induced by the Ar bubbles, as also observed experimentally. In the newly added Supplementary Fig. S6 the correlation between Ar bubble size and number of domain walls emerging from it is presented. The main text was also revised accordingly (see revised and extended section 'Origin of the Network').

In summary, please provide more atomic-level characterization on the structural origin of the 'strain' that the authors speak of here and how that translates to different types of domain wall network motif.

We have extended our discussion on the role of the Ar-bubbles for the generation of the domain wall network and more explicitly mentioned the results shown in Fig. S7 also in the main text (see revised and extended section 'Origin of the Network'). To clarify the relation between the Ar bubble and the selection of a specific orientational domain we have added a top view sketch in Fig. S7b. The sketch illustrates that the strain around the bubble leads to the characteristic alignment of the domains with respect to the junction core.

2. Does the domain wall actually 'map' the surface strain/distortion? If this is true, this could be very exciting as one would be able to deposit these Mn-film on semi-rigid materials (e.g. freestanding semiconducting 2D materials) and could follow in real-time the rippling of materials and hence, modulation of band gaps on various locales of the 2D material. In addition, one could also place such anti-ferromagnetic material on a substrate prepared with Si-towers to generate well-separated junctions, opening new avenues for condensed matter physics experiments. This however can only be revealed with further data analysis (see point 1). We thereby urge the authors to extract these conclusions from their present data and add, as an outlook, exciting future experiments enabled by this discovery.

We have shown that local strain can induce the formation of particular orientational domains in antiferromagnets. The mechanism for this is understood to be the exchange-interaction-induced shift between layers, which uniquely couples to the strain. In our case the magnetic domain walls are thus located at positions in the film where the shear changes its direction. Based on our findings for our model systems, we anticipate that all phenomena suggested by the reviewer are possible. However, the magnetic state of the film is governed by many aspects, e.g. symmetry, atomic distances, and hybridization at interfaces, so only experiments will be able to verify if this works for a particular system, or if maybe alternative antiferromagnetic materials might better reflect the strain in a given substrate. We expect that our findings will initiate further research in this area, along the lines of the reviewer's suggestion and added some sentences to the discussion section:

'... In turn antiferromagnetic thin films could be used as sensors of local strain at the surface of a sample of interest. Local strain in bulk antiferromagnets can lead to more complex three-dimensional domain wall networks. We anticipate that our proof-of-principle demonstration will trigger further investigations of the intimate relationship of structure and magnetism. ...'

3. It appears, from Fig. 2e, that the domain walls should meet exactly at some common point. However, Fig 2c shows that the domain walls do not exactly meet in the middle and form as suggested later in the text as a windmill motif. How do the authors reconcile the difference between the experimental data and the DFT results? Also see Figure 4d.

As the system is quite complex we try to guide the reader through the different aspects one by one. Figure 2e,f are spin dynamics simulations (not DFT results, for more details on spin dynamics simulation see methods section) of a symmetric unshifted monolayer, and serve to help the reader understand the spin texture at triple-junctions. Indeed, this is the spin texture that is expected for any system, shifted or not shifted, monolayer or bilayer, in the area where the three orientational domains of the RW-AFM state meet. The reviewer is correct that already in Fig. 2c,d one can see the asymmetry of the double-layer system under study here. However, we decided to guide the reader through the manuscript discussing one aspect at a time, leaving the discussion about the magnetic-exchange-driven shift to Fig. 4, after the origin has been revealed by DFT calculations in Fig. 3. We hope the reviewer will understand our decision to leave this unchanged. However, we have added one sentence in the methods section to prevent any misunderstandings. The additional sentence in the 'Methods: Spin models' section is marked in **bold** text below:

'...we have used a monolayer setup with generic parameters to illustrate the different building blocks for the network, without the aspiration to perfectly describe the system under study. **In particular, for this monolayer setup we also neglected the shift of the top layer with respect to the bottom layer.** To set up the system we have used the following Hamiltonian... '

4. It is unclear how the authors relate taking into account the breaking of the three-fold symmetry of the substrate by the RW-AFM state means that the Mn atoms can shift their positions (Lines 187-192). Can the authors clarify this link?

The expected structural symmetry of the paramagnetic system is C_3 (three-fold rotational axis perpendicular to the surface) with Mn atoms residing in hollow sites. However, the symmetry of the antiferromagnetic state is lower, and in the combined system of antiferromagnetic rows with Mn atoms residing in hollow sites only one mirror plane perpendicular to the stripes remains, see Fig. 3c, where the mirror plane includes the horizontal axis. Any further atom relaxations that maintain this symmetry are now allowed, including a shift of the top layer parallel to the remaining mirror plane.

We have added parts of this discussion in the context of Fig. 3, and now the text reads (new text indicated in **bold**):

‘However, we have not yet taken into account that the RW-AFM state breaks the three-fold symmetry of the substrate. **In the system displayed in Fig. 3c only one mirror plane perpendicular to the dotted line remains. As long as this symmetry is maintained** the Mn atoms may in principle shift out of the hollow sites, **which may lead to a lower total energy of this magnetic state [48].**’

5. For completeness, should the authors not show the calculated spin-averaged STM images based on the Tersoff-Hamann model for the RW-AFM state (unshifted), at least in the SI? Especially, if the authors are using the calculations to help substantiate the claim that the Mn atoms on the top layer is shifted. I do agree, however, that the features purported to be the top-layer Mn atoms (Figure 4c) are not aligned across the domain walls.

We thank the reviewer for pointing out that this data is missing. In the revised version of the Supplementary Information we have added Supplementary Fig. S4 which shows the calculated spin-averaged STM image for the RW-AFM state for the unshifted Mn double layer.

6. Given discussion in point 5, would this mean that the Mn double layer at elevated temperatures would host a different structure than the shifted top layer model proposed in this work? Since the origin of the shifted top layer is due to exchange interactions, presumably at higher temperatures, these interactions will play a minimal role (on average)?

Correct, we expect that in the paramagnetic state the Mn atoms reside in hollow sites.

On figure presentation,

7. Given the title and abstract of the paper, the manuscript will benefit from a new figure showing the relationship between Ar-bubble-induced surface distortion inducing the domain wall network. This figure should show two (or four) large-scale images of (a) the Mn film without surface/substrate distortion (including before- and after-film growth) and (b) the Mn film with surface/substrate distortion (again, including before- and after-film growth). We are aware that Fig. S1 somewhat accomplishes this, and we suggest to improve Fig. S1 (which is very hard to understand at the moment) and elevate Fig. S1 as the new Fig. 1 of the manuscript.

We prefer to keep the structure of the manuscript as is. Our main findings comprise strain-induced domain wall generation with specific junction types and connectivity, and non-coplanar triple- and hexa-junctions with topological orbital moments. We discuss the mechanism of strain-induced domain selection and alignment with respect to the junctions in the context of Supplementary Fig. S7. We consider the surface science related details of the distribution of Ar bubble size not particularly important for our main conclusions.

8. I am unsure as to what is being referred to in Fig. 1c as a hexagonal pattern (mentioned in Lines 106-107). Can the authors be clearer with it by including a guide on the figures? Especially if it appears as a hallmark of the 3Q state in SP-STM.

We have rephrased the indicated sentence to clarify this (**bold** indicates the revised text):

‘... **Towards** the center of this domain wall **the lines transition into a dotted pattern which locally has hexagonal symmetry. Thus** we conclude that they are **2Q**-superposition domain walls as previously observed for the RW-AFM state in the Mn monolayer on Re(0001) [11]. ...’

9. Please add an indication on Fig 1.c on which lines are being referred to in Lines 104-105: “The *lines* visible in the two orientational domains of Fig. 1c,d enclose an angle of 120° , and we find that all of the straight walls in the overview image (Fig. 1a,b) are of this type.”

We have rephrased the indicated sentence to clarify this (bold indicates the revised text):

‘... The RW-AFM state is identified in **SP-STM images** by its characteristic appearance of alternating bright and dark **lines, which correspond to atomic rows of opposite spin directions** [41, 42], see also **the to-scale** illustration of the magnetic state in Fig. 1d. The lines visible in the two orientational...’

On written presentation, we believe the authors will benefit from restructuring many of the sentences in this work to aid the reader in understanding the actual scientific value of the work rather than leaving the reader to decipher/guess the unclear/ambiguous sentences. There are many instances where a comma/punctuation should be introduced but were mistakenly left out. In addition, there are instances of ambiguous pronouns and ambiguous relative clauses in the manuscript. The authors should consider their text and rewrite to add clarity to their writing. Below are some examples:

We thank the reviewer for the thorough assessment of our manuscript and have carefully considered the suggestions below.

1. Missing comma in Line 14: “At the same time, ...”
2. Line 17: “... but [also] to exploit ...”
3. Please rewrite; wrong sentence structure in Line 20: “ranging from collinear magnetic domains, over non-collinear domain walls, to non-coplanar localized domain wall junctions”.
4. “for instance regarding” in Line 27 does not make sense. Replace with “such as”
5. Please rewrite. Incoherent sentencing in Lines 28-29: “Next to collinear antiferromagnets also systems with non-collinear magnetic order as in Mn₃Sn [4, 5] or non-coplanar order as in CoTa₃S₆ [6, 7] ...”.
6. Missing comma in Line 31: “At the same time, ...”
7. Line 33-34, rewrite to: “size of domains, the properties of antiferromagnetic domain walls are also relevant for spintronics applications”
8. Incoherent sentence. Please rewrite the sentence in lines 46 to 48.
9. Line 54: “In the case of ...”
10. Missing comma in Line 56: “Often, ...”
11. Incoherent sentence structure. Please rewrite the sentence in lines 60 to 63.
12. Missing comma in line 64: “In a spin spiral phase, ...”
13. typo in line 68: “A large-scale patterning ...”
14. Rephrase line 70: “... which manifest for instance in ... “ to “... which manifest in, for instance, ...”
15. . Please rewrite; wrong sentence structure in Line 77: “ranging from collinear magnetic domains, over non-collinear domain walls, to non-coplanar localized domain wall junctions”.
16. Missing comma in line 92: “In this system, ...”
17. Missing comma in line 98: “Here, ...”
18. Line 121: “adapt” instead of “adopt”?
19. Line 144: “Note, that within all DFT, ...”
20. Line 243: “After the analysis of the magnetic state in the triple-junctions we turn, ...”

Reviewer #2 (Remarks to the Author):

Reviewer #3 (Remarks to the Author):

The authors studied the spin configurations of double-layer Mn using SP-STM. According to the symmetry of the material, there are six different collinear ground states. Their main findings are as follows. First, the collinear ground states involve the lateral shift of $\sim 1 \text{ \AA}$ of Mn atoms in one of the two layers, verified by the DFT calculations. Second, two ground states meet along a domain wall, across which the spin configurations change in a co-planar fashion. The displacement vector of the moved Mn atoms changes across the domain wall too. Thirdly, three ground states meet at one junction point, around which the spin structures are non-coplanar. In this case, the three displacement vectors of the involved three ground states rotate around the junction point, showing the specific structural chirality. Overall, they found that domain walls can form a network with multiple junctions with characteristic structural chirality, which they called 'strain-driven domain wall network with chiral junctions.' While it is well-known that domain walls can occur naturally in various magnetic materials, the question of whether domain walls can form a network in magnetic systems has been studied little despite its fundamental interest as well as technological relevance. The current manuscript reports clear evidence of domain-wall networks with rich topological characteristics in antiferromagnets, which will advance our fundamental understanding of magnetism. It has also potential utility in spintronics as a platform over which spin information can flow. For this reason, in principle, I can recommend the publication of the manuscript in Nature Communications, provided that the following comments can be addressed.

We thank the referee for carefully reviewing our work, the positive evaluation of our manuscript, and for pointing out its relevance for researchers working in magnetism and spintronics. Based on the reviewer's valuable comments and suggestions we have further improved our manuscript.

1) While the authors focused on the chirality of the Mn-atom displacement vectors at the junctions, the spin configurations around the triple junctions can be considered as half-skyrmions or merons of the Neel-order parameter, which can be characterized by the topological winding number (see, e.g., [Yu et al., "Transformation between meron and skyrmion topological spin textures in a chiral magnet," Nature 564, 95 (2018); Jane et al., "Antiferromagnetic half-skyrmions and bimerons at room temperature," Nature 590, 74 (2021)]). In the current manuscript, the triple junctions are characterized only by the geometric Y/T type and by the patterns of the structural displacement vectors. It would be instructive to characterize triple junctions by their winding numbers, which are rooted in spin degrees of freedom.

We are reluctant to use the spin degree of freedom for the characterization of the junction as the spin texture changes from 1Q (RW-AFM) in the domains to 3Q at the junction, making a coherent definition difficult. In particular in our opinion this also prevents an analogy to merons because in our case the core exhibits a different type of spin structure compared to the outside.

Instead, to connect to the next point of the reviewer, in the new version we now use the different helicity of the shift directions for the different triple junctions to guide the reader through the aspect of structural chirality (see last paragraph of section 'Characterization and Chirality of the Junctions').

2) The reason why Y and T types always occur together can be understood by the vortex-antivortex interaction in the XY systems. From Fig. 5, around the Y and Y* junction, the displacement vectors (blue arrows) rotate counterclockwise as we encircle the junction counterclockwise. On the contrary, around the T and T* junctions, the displacement vectors rotate clockwise as we encircle the junction counterclockwise. This indicates that Y/Y* junctions are vortices, whereas the T/T* junctions are antivortices in the analog of the XY model. The observation that Y types and T types are generated together can be explained easily by the well-known pairwise generation of vortices and antivortices in the XY model. I suggest that the authors incorporate this analogy in the manuscript to facilitate readers' understanding.

As discussed in our answer to the previous point in the revised version of the manuscript we use the helicity of the shift directions to discuss the structural chirality. Accordingly, we explain the connectivity based on the helicity (see last paragraph of section 'Characterization and Chirality of the Junctions').

3) The authors calculated the topological orbital moment (TOM) for one type of hexa-junctions. However, TOM is expected to be finite for all types of triple junctions as well as all types of hexa-junctions. Furthermore, TOM should depend on the type of junctions, such as Y, Y*, T, T* type for triple junctions. Can the authors show the TOM calculation results for all different types of triple junctions and hexa-junctions? This will enable distinguishing different types of junctions.

DFT calculations demonstrate that the TOM per magnetic unit cell of the 3Q state for the non-sheared Mn double layer on Ir(111) is about $0.26 \mu_B$. The total TOM for a specific junction is then given by the number of 3Q unit cells formed in the vicinity of that junction. We have not calculated by DFT any of the junctions, as this is computationally prohibitive even on supercomputers due to the large structures of the junctions and the need to consider non-coplanar (non-collinear) magnetic states. Experimentally we find that the area of the hexagonal pattern indicative of the 3Q state is in general largest for the hexa-junctions, whereas the 3Q area in the triple-junctions is smaller. This suggests that the hexa-junctions can give rise to the largest total TOM. According to our spin dynamics simulations this is expected for the symmetric hexa-junction type as shown in Fig. 6c, whereas in other hexa-junctions as displayed in Supplementary Fig. S5e,f,g the TOM may cancel.

We have added some of this information in the sections "Origin of the network" and "Emergent Topological properties of the junctions" (bold indicates the revised text):

'... The experimentally observed offset of the wall positions with respect to the center of the junction reflects the strain due to this shearing of the Mn film. **The symmetric arrangement of shear directions around the junction suggests a non-sheared Mn double layer in the Ar bubble center.'**

'..For both types of the bilayer 3Q state we have calculated the TOM for each **Mn atom of the non-unsheared Mn double layer on Ir(111)** via DFT.'

'... all the triple- and hexa-junctions exhibit the non-coplanar 3Q state and therefore also an associated local TOM. **The size of the total TOM is given by the area where the spin texture is non-coplanar. The experiments show that the expected TOM is largest for the highly symmetric hexa-junction, and from the TOM per magnetic unit cell and an estimated area of 20 nm^2 we derive a total TOM of about $20 \mu_B$ for the hexa-junction shown in Fig. 6a,b.'**

4) Please provide the 'color scheme' for spin configurations such as Fig. 2e and Fig. 6c.

We thank the reviewer for pointing out that this information is missing and we have added it to the respective captions.

Reviewer #4 (Remarks to the Author):

The manuscript presents a high-quality study that combines spin-polarized scanning tunneling microscopy and first-principles calculations to investigate a strain-induced domain wall network in a Mn double-layer on Ir(111). The authors convincingly identify a range of magnetic building blocks, including collinear antiferromagnetic domains, non-collinear domain walls, and non-coplanar 3Q-type triple junctions. The discovery of chiral junctions exhibiting topological orbital magnetization is particularly compelling and provides valuable insight into the physics of emergent topological spin textures in antiferromagnets.

I regard this work as scientifically rigorous and potentially impactful, especially in the areas of topological magnetism and emergent electronic phenomena. The identification of chiral triple junctions hosting non-coplanar 3Q spin textures and associated topological orbital magnetization represents a novel and significant result that will be of strong interest to the fundamental physics community.

However, I do not believe the manuscript fully aligns with the scope of this journal. Its emphasis is primarily on fundamental physical mechanisms, and its appeal may be more limited in terms of broad, interdisciplinary readership. For these reasons, I recommend the authors consider submitting their work to Communications Physics, where the journal's focus on foundational physical insights and emergent topological effects is likely to be a better match.

We thank the reviewer for evaluating our manuscript and for the positive comments, in particular the statement that our results are novel, significant, and interesting for the fundamental physics community. On the contrary with the reviewer's judgement we believe that this perfectly aligns with the scope of Nature Communications, which has published many papers that provide deep insight into fundamental physics.

General comments:

1. The study is performed at ultra-low temperature ($T=4$ K). While this enables high-resolution imaging, it raises questions about the relevance and stability of the observed domain wall network and 3Q states at technologically relevant or even intermediate temperatures. The lack of any temperature-dependent study limits the broader applicability of the findings.

This study serves as a proof of principle study on a model-type system which allows atomic-scale imaging of all constituents of the domain wall network using spin-polarized STM. We anticipate that strain-induced generation of antiferromagnetic domain walls may be applicable to several other sample systems.

2. Although the manuscript touches on potential transport phenomena (topological Hall effects), no experimental verification of these properties is provided. This weakens the impact for journals with an application-oriented or broader audience.

In our work, we demonstrate the creation of a domain-wall network with topological properties in an antiferromagnetic material. Since we study a thin film at a surface and use cutting-edge SP-STM, we can resolve all of the intriguing spin structures in the junctions of the network down to the atomic scale and relate them to structural properties such as local strain and shearing. For transport measurements, however, different types of samples, e.g. layered or bulk-like materials, would be beneficial. Therefore,

such studies are complementary to ours. We expect that our work on a model-type system will trigger future experiments on bulk-like materials which can make use of the anticipated topological transport properties.

Note that in the past there are many examples in which experimental work on model systems such as ours provided the first microscopic insight into novel magnetic phenomena before application-oriented research was possible to exploit them. For example, the interfacial Dzyaloshinskii-Moriya interaction was discovered in a Mn monolayer on W(110) single crystal surface (Bode *et al.*, Nature (2007)) and later used for current-driven chiral domain wall motion (Emori *et al.*, Nat. Mat. (2013)). Skyrmions at transition-metal interfaces were also first discovered at single crystal surfaces (Romming *et al.*, Science (2013) and Heinze *et al.*, Nat. Phys. (2011)) before being applied in transition-metal multilayers (e.g. Moreau-Luchaire *et al.*, Nat. Nanotechnol. (2015)). Another example is the triple-Q state predicted more than 20 years ago (Kurz *et al.* PRL (2001)) and experimentally first observed in an ultrathin film on a single crystal surface (Spethmann *et al.*, PRL (2020)), while the spontaneous Hall effect due to the triple-Q state was revealed in intercalated van der Waals materials (Park *et al.*, Nat. Commun. (2023), Takagi *et al.*, Nat. Phys. (2023)).

3. The introduction, while thorough, can benefit from a more focused motivation paragraph early on to guide readers unfamiliar with the domain.

In the revised version we have added a paragraph on the properties and formation of superposition states to guide readers unfamiliar with this concept.

4. Some sections could be shortened or streamlined, especially those reviewing prior work, this could help draw focus to the new contributions.

We have considered this comment, however, we consider the prior work reviewed in our introduction relevant for the context of our work, so we were reluctant to shorten it.

5. The energetic contradiction between DFT predicting the 3Q state as lowest-energy and experimental observation of RW-AFM domains is well-resolved by considering structural relaxation. However, the paper could do more to explain this transition physically, especially for non-specialists.

We have extended our discussion on the higher-order interactions in the revised version of the manuscript (lines 158-162) in order to clarify why these terms are typically small. In the context of the transition to the RW-AFM state due to the shift of the Mn top layer we further point out the competition of pair-wise and higher-order exchange (lines 206-208). In the revised manuscript, we have also added a reference (lines 171-172) to a manuscript recently submitted to PRB in which we present the DFT calculations of spin spirals for the pseudomorphic (i.e. unshifted) Mn double layer on Ir(111) (B. Beyer *et al.*, arxiv:2506:05091). In that work, we have determined the inter- and intralayer pair-wise exchange constants explicitly. Further, we have extended the atomistic spin model and show explicitly that the 3Q state is stabilized by interlayer higher-order exchange (lines 183-184).

6. While the paper ends with strong conclusions, it would benefit from a brief discussion of experimental limitations, for example the field, temperature, thickness dependence.

Our work serves as a proof-of-principle on a model-type system. Future studies will show how this can be transferred to other systems of interest. We have added this to the discussion section.

Minor comments:

1. Figure references sometimes appear before the figure is described in sufficient detail.

We are unsure what the reviewer is referring to.

2. Add clear units to colorbars and vertical axes in DFT energy plots.

The color scales of the topographic images and the current images are provided in the respective captions (we have added that of Fig. 4c, which was missing). We are unsure which vertical axes of the DFT energy plots are unclear.

We thank the reviewer for the general as well as the minor comments and have considered all of them carefully.

REVIEWER COMMENTS

Reviewer #1 (Remarks to the Author):

Note: This review includes comments from another co-reviewer as a part of Nature Communications initiative facilitating training in peer review.

We thank the authors in implementing some of our suggestions, which we think have clarified the main findings of the paper. We congratulate the authors for this discovery and we hereby endorse the acceptance and publication of the manuscript in Nature Communications.

We thank the reviewer for endorsing the acceptance and publication in Nature Communications.

Reviewer #2 (Remarks to the Author):

We thank the early career reviewer for contributing to this review process and the recommendation of the acceptance and publication of our manuscript in Nature Communications.

Reviewer #3 (Remarks to the Author):

The authors have responded well to the comments of the reviewers, including myself, leading to the recommendation of the publication of the manuscript in Nature Communications.

We thank the reviewer for emphasizing that we have responded well to the comments of all reviewers, and the conclusion to recommend publication in Nature Communications.

Reviewer #4 (Remarks to the Author):

After reviewing the author's rebuttal letter and the revised manuscript, I would like to reiterate and clarify my concerns:

My initial comment addressed the lack of finite temperature measurements, as the entire study is conducted at 4K. The authors did not provide a clear explanation for why this temperature was chosen, nor did they discuss the potential behavior of the strain-induced domain wall network at higher temperatures. I am particularly interested in whether this domain wall network persists at larger temperatures. While it is valid to conduct a fundamental study at very low temperatures, the authors must clarify that the scope of this work is limited to 4K and does not necessarily have direct applications at higher temperatures. Their response to this comment was unsatisfactory, as they failed to explain the rationale for using 4K or provide any discussion of how the results might change at different temperatures. A clearer justification is needed, especially if the study aims to contribute to applications in a broader temperature range.

We thank the reviewer for again evaluating our manuscript. The experimental study was conducted at 4.2 K because the method of spin-polarized STM, which we use to uncover the spin texture of the domain, the domain walls, and the domain wall junctions down to the atomic scale, is very challenging and the low temperature facilitates a stable tip magnetization and a high junction stability. SP-STM measurements at higher temperatures become increasingly challenging. As mentioned in our previous response our work serves as a proof-of-principle, and we expect that such domain wall networks can be generated in many different antiferromagnetic materials.

During the initial stages of the measurements we have experimented with local stacking faults for the generation of antiferromagnetic domain walls. Such measurements were also done at 80 K, and we have now added another supplementary figure to show these results. Furthermore we have modified the following paragraph in the main text (first paragraph of the Discussion section) to now read the following:

“... The local strain induced by the Ar bubbles is in turn responsible for the preference of specific orientations of the adjacent domains. **While also other sources of local strain, such as defect lines (see Fig. S8), can induce domain wall networks, for our system the Ar bubbles lead to the best connectivity.** Our DFT calculations have confirmed that compressive strain at the rim of the Ar bubbles selects one out of the three possible orientational domains. This can be viewed as a reciprocal effect of the magnetism-driven shearing of the magnetic film. **Because this magnetism-driven shearing within the antiferromagnet is the origin of the strain-induced network, we expect that the network exists up to the magnetic ordering temperature. While for our system we expect that the ordering temperature is much higher as the roughly 100 K reported for a similar system with only one monolayer [59], we have examined the existence of a domain wall network only up to 80 K (see Fig. S8). ...**”

Additionally, the authors did not adequately address comments 4 and 6 from my previous review. Their replies to these points were either insufficient or absent. These comments were central to the clarity and depth of the manuscript, and a more thorough response is required to address the underlying issues.

In the previous report, comments 4 and 6 were merely phrased as suggestions (‘4. Some sections could be shortened or streamlined’ and ‘6. ... the paper ... would benefit from ... ’), and we cannot understand why now the Reviewer states that ‘these comments are central to the clarity and the depth of the manuscript.’

We would like to emphasize that we had addressed these points in the previous round. However, we arrived at the conclusion that a shortening of sections is not possible due to the complexity and novelty of our results, and gave priority to a thorough discussion of all points for the benefit of the reader. Furthermore, as we consider our work as a proof of principle study, we did not further discuss the experimental limitations specific to this system. In the revised version we have now more explicitly explained that we expect structurally driven domain wall networks to be stable up to the critical temperature of the magnetic order, and we have included the new Fig. S8 with experimental data at 80 K.